# South Atlantic overturning and heat transport variations in an ocean reanalysis ensemble and observation-based estimates

Jonathan Andrew Baker[1], Richard Renshaw[1], Laura Claire Jackson[1], Clotilde

5    Dubois[2], Doroteaciro Iovino[3], Hao Zuo[4], Renellys C. Perez[5], Shenfu Dong[5], Marion Kersalé[6], Michael Mayer[7,4], Johannes Mayer[7], Sabrina Speich[8], Tarron Lamont[9,10,11]

1: Met Office, UK; 2: Mercator Ocean International, France; 3: Centro Euro-Mediterraneo sui Cambiamenti Climatici, Italy; 4: European Centre for Medium-Range Weather Forecasts, UK; 5: National Oceanic and Atmospheric Administration/Atlantic Oceanographic and Meteorological Laboratory, USA; 6: Direction

10   Générale de l'Armement, Ingénierie des projets, Paris, France 7: University of Vienna, Austria; 8: Laboratoire de Météorologie Dynamique–IPSL, Ecole Normale Supérieure, France; 9: University of Cape Town, South Africa, 10: Department of Environment, Forestry and Fisheries, Oceans and Coasts Research Branch, South Africa, 11: Bayworld Centre for Research and Education, South Africa

*Correspondence to*: Jonathan Baker (jonathan.baker@metoffice.gov.uk)

**Abstract.** The variability of the South Atlantic meridional overturning circulation and meridional heat transport measured across 34.5°S during 2013–2017 differs significantly between observational and ocean reanalysis estimates. Variability in an ocean reanalysis ensemble and an eddy-resolving reanalysis is similar to an altimeter-based estimate, but smaller than energy-budget and mooring-based estimates. Over 1993–2020, there is no long-term trend in the ensemble-mean overturning and heat transport, although there are inter-model differences, whereas the altimeter-based and energy-budget estimate transports increase over this period. Time-mean overturning volume transport (and the depth of maximum overturning) across 34.5°S in the ensemble and observations are similar, whereas the corresponding mean heat transports differ by up to 0.3 PW. The seasonal cycle of these transports varies between estimates, due to differences in the methods for estimating the geostrophic flow and the sampling characteristics of the observational approaches. The baroclinic, barotropic and Ekman MOC components tend to augment each other in mooring-based estimates, whereas in other estimates they tend to oppose each other so the monthly-mean, inter-annual and seasonal MOC anomalies have a greater magnitude in the mooring-based estimates. Thus, the mean and variation of real world South Atlantic transports, and the amplitude of their fluctuations, are still uncertain. Ocean reanalyses are useful tools to identify and understand the source of these differences and the mechanisms that control volume and heat transport variability in the South Atlantic, a region critical for determining the global overturning pathways and inter-basin transports.

**Short Summary.** We use ocean reanalyses, in which ocean models are combined with observations, to infer past changes in ocean circulation and heat transport in the South Atlantic. Comparing these estimates with other observation-based estimates, we find differences in their trends, variability, and mean heat transport, but closer agreement in their mean overturning strength. Ocean reanalyses can help us understand the cause of these differences, which could improve estimates of ocean transports in this region.

## 1 Introduction

The Meridional Overturning Circulation (MOC) modulates climate on seasonal to millennial timescales via its meridional transport of freshwater, heat and carbon through the global ocean (Rahmstorf, 2015; Weijer et al., 2019; Buckley and Marshall, 2016). It is therefore important to understand how the Atlantic MOC (AMOC), which dominates the upper cell of the global MOC, is changing. Changes in overturning in the South Atlantic are particularly important because they play a crucial role in determining the pathways of the global overturning circulation (Baker et al., 2021, 2020; Xu et al., 2022; Nadeau and Jansen, 2020), while freshwater transports in the South Atlantic impact the stability of the AMOC (Garzoli and Matano, 2011; Hawkins et al., 2011; Weijer et al., 2019, 2002). Transport changes here could determine the rate at which the AMOC weakens in response to increased greenhouse gas emissions (Weijer et al., 2020; Collins et al. 2019), beyond the weakening that may have already occurred over the past century (Caesar et al., 2018; Rahmstorf, 2015; Thornalley et al., 2018).

From September 2013 to July 2017, the expanded nine-site South Atlantic Meridional Overturning Circulation – Basin-wide Array (SAMBA) (Fig. 1) has collected measurements from which both daily meridional transports of heat and volume across 34.5°S can be estimated (Kersalé et al., 2020, 2021). Volume transports were also estimated during 2009-2010 using the two-site pilot configuration of the SAMBA array (Meinen et al., 2018, 2013). These studies have improved our understanding of the variability of the overturning circulation and

meridional heat transport (MHT) in this region. The SAMBA array has improved mooring coverage since 2021 (Chidichimo et al., 2023), but data recorded after 2017 have yet to be incorporated into published AMOC or MHT estimates.

Since MOC and MHT estimates are currently only available from SAMBA during 2013–2017, longer-term variations must be inferred using model- and alternative observation-based estimates (Garzoli et al., 2013; Goes et al., 2015; Dong et al., 2009; Mignac et al., 2018; Biastoch et al., 2021; Caínzos et al., 2022). This includes transport estimates derived from satellite sea level anomalies (SLA) and in-situ data (Dong et al., 2015; Majumder et al., 2016). Although Majumder et al., (2016) found large differences between ocean reanalyses and their observation-based estimate from 2000–2014, ocean reanalyses agree better with observations than free-running models (Mignac et al., 2018). Dong et al. (2021) generated MOC and MHT estimates over 1993–2021 from a synthetic method combining in-situ and satellite data (updated from Dong et al., 2015) that agreed well with XBT-derived MOC and MHT estimates in the South Atlantic. The MHT estimates from Dong et al. (2021), however, differed significantly from energy-budget MHT estimates produced by Trenberth et al. (2019). All of the aforementioned transport estimates vary less than the nine-site SAMBA array estimates (Kersalé et al., 2021, 2020).

We aim to build upon these studies by comparing an ensemble of global ocean reanalyses (product ref's 1, 2, 3) directly against the observation-based estimates available over the SAMBA (2013–2017) and the altimetry (1993–2020) time periods. We also compare the reanalyses with new energy-budget MHT estimates at 34.5°S, which are analogous to an estimate at 26°N in the North Atlantic of Mayer et al. (2022), that is well correlated with observed transports across the RAPID array. While SAMBA array studies have primarily focused on daily-to-seasonal variability; here we focus on monthly-to-interannual variability. All of the time series were averaged to represent monthly values prior to further analysis.

Ocean reanalyses may provide realistic three-dimensional estimates of past changes in the South Atlantic overturning and heat transport (Mignac et al., 2018), and thus could be a useful tool to infer the nature and cause of past MOC and MHT variability. An earlier version of the reanalysis ensemble used in this study provides a good representation of the subtropical and subpolar North Atlantic overturning circulation (Jackson et al., 2018; Jackson et al., 2019; Baker et al., 2022); thus, it may also accurately simulate changes in the South Atlantic.

**2 Data and Methods**

**2.1 Data**

We use an ensemble of eddy-permitting (¼ degree horizontal resolution) global ocean reanalyses. These are GloRanV14 (an improvement of GloSea5, MacLachlan et al., (2015)), C-GLORSv7 (Storto et al., 2016), GLORYS2V4 (Lellouche et al., 2013), and ORAP6 (Zuo et al., 2021). Together, these four reanalyses form a new Copernicus Marine Environment Monitoring Service (CMEMS) reanalyses ensemble, updating product ref 1 (see Table 1). We also use an eddy-resolving ($^1/_{12}$ degree) global ocean reanalysis, GLORYS12V1 (product ref 4). Each reanalysis uses the NEMO ocean model, but the sea-ice model and data assimilation techniques differ. Each

reanalysis is constrained by observations and is driven by atmospheric forcing from either ERA5 (Hersbach et al., 2020) or ERA-Interim (Dee et al., 2011) over the period 1993–2020, with GloRan extended to December 2021. They all assimilate satellite SLA, sea-ice concentrations, and in-situ temperature and salinity, and they either assimilate satellite sea surface temperature (SST) or implement SST nudging.

We compare the MOC and MHT from the ensemble with the SAMBA-based estimates of Kersalé et al. (2020; 2021), the altimeter-based estimate of Dong et al. (2021), and the energy-budget MHT estimates of Trenberth et al. (2019) and Mayer et al. (2022).

The energy-budget estimates of Mayer et al. (2022) calculate the net surface heat flux using top-of-atmosphere radiative fluxes from CERES-EBAF (Loeb et al., 2018) with a backward extension (Liu et al., 2020), and atmospheric energy budget quantities from ERA5 (see Mayer et al., 2021 for methods). These are combined with ocean heat content (OHC) tendencies from ocean reanalyses to infer the MHT. Mayer et al. (2022) use OHC tendencies from ORAP6 ("Mayer_ORAP6" in figures); here we use an additional (unpublished) ORAS5-based estimate ("Mayer_ORAS5"), using OHC tendencies from ORAS5 (Zuo et al., 2019), the same as that used in the Trenberth et al. (2019) estimate. For further details, see the supplementary materials. We note that energy-budget estimates may accumulate errors at southern latitudes, since they are integrated southward from high, northern latitudes (Dong et al., 2021).

## 2.2 Methods

Ensemble-mean and spread, and the time-mean of the altimeter-based and Mayer energy-budget estimates are calculated over 1993–2020 and over the 2013–2017 SAMBA observational period. We calculate monthly-mean MOC across 34.5°S in depth coordinates, using commonly applied methods (e.g., Frajka-Williams et al., 2019), integrating monthly-mean velocity from coast-to-coast and from the surface down to the seafloor with a zero-net-volume transport constraint applied. Without this constraint, the ensemble-mean has a net southward transport through the section over the observational period of 1.14 Sv (as do the individual reanalyses), and GLORYS12V1 has a net southward transport of 3.1 Sv, but the constraint has only a small impact on MOC estimates (Table 1). For the reanalysis, the MHT is calculated by integrating the product of monthly-mean model velocity and temperature (scaled by density and specific heat coefficient) across the whole section with a zero-net-volume transport constraint applied. Each observational product applies its own constraint to reference the flow due to differences in their geostrophic techniques. The altimeter-based dataset references the flow to the time-mean YoMaHA velocities at 1000 m (Katsumata & Yoshinari, 2010; Lebedev et al., 2007) and uses a zero net mass transport constraint (Dong et al., 2021). Kersalé et al. (2020) use models to reference the time-mean barotropic component at 1500 db, and bottom pressure measurements from the moorings provide the time-varying barotropic velocity component.

We calculate the overturning profiles, the monthly-to-interannual variation, and the seasonal cycles of the upper cell MOC and the total MHT in each dataset. We separate the transports into their Ekman and geostrophic components. In the reanalyses, the Ekman component is calculated using the ERA5 or ERA-Interim wind stress, and for MHT, the zonal-mean SST across the section, assuming SST is representative of the Ekman layer temperature. The geostrophic component is calculated as a residual of the total and Ekman transports.

We also calculate the baroclinic and barotropic components of the ensemble's geostrophic MOC. We use thermal wind balance and the model's geopotential height anomalies to estimate the baroclinic velocities (e.g., see Perez et al., 2011), integrating these from the deep ocean to the surface. The reference level is set ~1000 m above the ocean floor, above the unphysically large zonal gradients in geopotential height anomaly that exist in the deepest layers of the model. Thus, the reference level depth varies spatially (~2000 m to ~4000 m deep) due to the bathymetry, but it is constant in time. The reference velocity is not required to calculate the baroclinic MOC anomalies, so we set the baroclinic velocity to zero at the reference level depth. A visual inspection confirmed that the large month-to-month spatial variations in the baroclinic velocity field are in good agreement with the associated changes in the total velocity field. We tested the method using different reference level depths that generated similar monthly-mean MOC anomalies (not shown). We calculate the baroclinic component of the MOC by integrating the baroclinic velocities from the surface down to the depth of the time-mean total MOC maximum in each reanalysis (~1250 m over 2013–2017). We calculate the barotropic component as a residual of the geostrophic and baroclinic MOC anomalies. The baroclinic and barotropic MOC anomalies in the reanalyses and in SAMBA estimates are not directly comparable because the reference levels differ. However, our baroclinic MOC anomaly estimate in the reanalyses accounts for baroclinic velocity variations from around 1000 m above the ocean floor to the surface over which the velocities are greatest and have large monthly variation.

## 3 Results

### 3.1 MOC Profiles and statistics of variability

The ensemble of reanalyses captures the main structure of the observed overturning profile (Fig. 1a). The depth and strength of the maximum overturning is similar among all estimates with a range of ~15-18 Sv (Fig. 1a). The profiles diverge in the deeper ocean, with a weaker than observed lower overturning cell and southward flow in the ensemble (i.e., the MOC decreases more gradually with depth). The reanalyses are less accurate at depth due to there being fewer observations to constrain the flow. The overturning profiles of the ensemble and GLORYS12V1 in density space have no negative transport (i.e., no abyssal cell), and their MOC is stronger than in depth space (Fig. 1b). The temporal variability of their upper MOC strength at 34.5°S, however, is fairly insensitive to the vertical coordinate system used for integration (Fig. S1). We therefore focus on the MOC in depth space because the reanalyses can then be directly compared with the observational estimates.

We analyse the basic statistics of the variability of the maximum MOC strength and the MHT by looking at their time-mean and standard deviation over 2013–2017 and 1993–2020. The time-mean MOC estimates have a range of 15.5–18.7 Sv, with the ensemble-mean (labelled "mean" in figures) being only slightly weaker than the altimeter-based estimate and that observed across SAMBA (crosses in Fig. 2a). The time-mean MHT estimates have a range of 0.31–0.61 PW (crosses in Fig. 2c). Relative to the ensemble-mean values (MOC: 16.56 Sv; MHT: 0.36 PW), the time-mean MHT range has a 75% increase from its minimum to maximum value (excluding the energy-budget estimates) compared to only a 20% increase for the time-mean MOC range. These ranges are within the documented uncertainty of SAMBA (Table 2). The ensemble-mean MHT is similar to the energy-budget estimates based on Mayer et al. (2022) (Fig. 2c). While there is inter-model spread in the ensemble time-mean transports (crosses in Fig. 2b,d), the spread is smaller than the uncertainty in SAMBA (Table 2), although it is more comparable for the MHT than for the MOC.

Monthly-mean variability (i.e., the standard deviation) of MOC and MHT in the ensemble is similar to the altimeter-based estimate over 2013–2017 and 1993–2020, whereas variability observed from SAMBA is much greater (Fig's 2a,c and 3a-d, and Table 2), with significant differences ($p < 0.05$ in an F-test for equality of two variances). Similarly, the ensemble-mean timeseries is significantly ($p < 0.05$) correlated with the altimeter-based estimate (r=0.63 for MOC; r=0.77 for MHT, over 2013–2017), but it is not well correlated with SAMBA ($r < 0.1$). The monthly-mean SAMBA estimates (Fig. 3a,b) and the Mayer energy-budget MHT estimates have high-frequency variations of comparable magnitude (Fig. 3b,d and Table 2), although their variability is uncorrelated. "Mayer_ORAP6" is weakly correlated with the ensemble-mean and altimeter-based estimate (r=0.14, r=0.19 over 1993–2017; r=0.28, r=0.32 over 2013–2017, for the respective datasets). "Mayer_ORAS5" has a higher correlation with the ensemble-mean and altimeter-based estimate (r=0.30, r=0.32 over 1993–2017; r=0.52, r=0.57 over 2013–2017, for the respective datasets). The GloRan reanalysis run with and without assimilating altimetry data (not shown) has a similar correlation with the altimeter-based estimate (r=0.52 vs r=0.56 for MOC over 2013–2017). Thus, the strong correlation between ensemble-mean and altimeter-based estimates is not dependent on directly assimilating altimetry data. The experimental reanalysis does, however, still assimilate in-situ and satellite temperature and salinity data, which would serve to constrain thermosteric and halosteric, respectively, contributions to sea level. In the 12-month running mean estimates (Fig. 3e,f), the ensemble-mean is only weakly correlated with the altimeter-based estimate (r=0.24 for MOC; r=0.25 for MHT), so their high monthly-mean correlation is largely due to similar seasonal variability.

The GLORYS12V1 reanalysis has a larger time-mean MOC and MHT than the ensemble-mean (and GLORYS2V4). It has similar monthly-mean variability to the lower resolution reanalyses, slightly larger than the ensemble-mean, but smaller than GLORYS2V4 (Table 2). It is also significantly correlated with the ensemble-mean (r=0.80 for MOC; r=0.84 for MHT, over 1993–2019). Thus, fully resolving (as opposed to only permitting) eddies in the ocean reanalyses considered here is important to infer the time-mean transports across 34.5°S, but has minimal impact on the variation of the monthly-mean transports.

The 12-month running mean MOC and MHT in the ensemble over 1993–2020 are relatively stable (Fig. 3e,f), with similar ensemble-mean values to those during 2013–2017 (Table 2) and no significant trend over 1993–2020. However, the individual reanalyses have significant ($p < 0.05$) trends in the MOC over 1993–2020 with differing sign and magnitude (Table 2). In contrast, only GloRan has a significant (increasing) trend in MHT (~0.042 PW/decade). GLORYS12V1 has no significant trend in MOC or MHT. Hence, there is uncertainty in the long-term trends amongst the reanalyses.

The altimeter-based estimate has significant ($p < 0.05$) increases in MOC (~0.66 Sv/decade) and MHT (~0.036 PW/decade) over 1993–2020. The aforementioned MHT trends are similar over 1993–2016 (GloRanV14: ~0.047 PW/decade; altimeter: ~0.032 PW/decade). There is a significant increase in MHT over 1993–2016 in both the ORAS5- (~0.086 PW/decade) and ORAP6- (~0.094 PW/decade) based Mayer estimates. The Trenberth estimate has a significant but weak decline (~-0.010 PW/decade) over 2000–2016; the Mayer estimates also declines over this period, but the trend is insignificant.

The 12-month running mean from SAMBA is entirely different to other estimates (Fig. 3e,f), with a rapid increase
in the MOC (~14 Sv) and MHT (~0.7 PW) from March 2014 to June 2016, followed by a rapid decline. Although
an extended timeseries is needed to determine longer timescale variations, the inter-annual variability captured by
SAMBA over 2013–2017 exceeds that of other estimates. Only the Mayer MHT estimates have inter-annual
variations of comparable magnitude, but those variations occur before 2013 (Fig. 3f).

## 3.2 Seasonal Cycles

There is a predominantly annual cycle in the ensemble-mean and altimeter-based transports, unlike the SAMBA
seasonal cycle that has a stronger semi-annual variability (Fig. 3c,d). While we show the ensemble-mean and
altimeter-based seasonal cycles over 2013–2017 (Fig. 4), the seasonal cycles derived over the full record lengths
are similar (not shown). The ensemble and altimeter-based overturning are weakest in austral summer, but the
ensemble is strongest in May/June, peaking two months after the altimeter-based estimate (Fig. 4, upper panels).
In contrast, the MOC in SAMBA is dominated by a semi-annual signal, with minima in April and September, and
maxima in August and December. There are year-to-year variations in the annual cycles of all estimates (not
shown), with variations in phase, shape and magnitude. In SAMBA, four years of observations are not long enough
to examine the sensitivity of the seasonal cycle to changing the time period, but given the strong high-frequency
variations, the seasonal cycle based on four years of data is unlikely to be robust.

The shape of the seasonal cycle in MHT is similar to that of the MOC for each estimate as expected given the
high correlation between the monthly-mean MHT and MOC (r=0.90, r=0.91, r=0.96 for ensemble-mean,
altimeter-based estimate, and SAMBA respectively over 2013–2017). The Mayer energy-budget estimates have
seasonal cycles dominated by an annual signal, with a larger magnitude range than other estimates. They are
similar to the Trenberth estimate, but with greater month-to-month variability. However, when averaged over the
2000–2016 period used in the Trenberth estimate rather than 2013–2017, they become smoother and closer to the
ensemble ("Mayer_ORAS5_2000–16" in Fig. 4).

Year-to-year variations in the annual cycles of each estimate over 2013–2017 (not shown), and differences in the
climatological seasonal cycle between each estimate (Fig. 4), stem from their geostrophic differences (Fig. 4,
lower panels), because the Ekman annual cycles are similar year-to-year (not shown) and for all estimates (Fig.
4, middle panels). Differences between estimates are clearer in the geostrophic component, peaking before the
ensemble-mean in the altimeter-based estimate and after the ensemble-mean in SAMBA. Thus, the Ekman and
geostrophic components tend to oppose each other in the altimeter-based estimate and augment each other in
SAMBA. This causes a greater increase in the magnitude of the total MOC and MHT seasonal cycles (relative to
their geostrophic components) in SAMBA than it does in the altimeter-based estimate, but a greater change in the
seasonal cycle phase and shape in the altimeter-based estimate (cf. Fig. 4, lower and upper panels). The relative
contribution of the Ekman component to the total MOC and MHT in the ensemble is nonetheless significantly
greater than in SAMBA. In the ensemble-mean (and in GLORYS12V1 and SAMBA), the geostrophic component
of the MOC (Fig. 4, lower left panel) has a second peak in November or December (i.e., austral spring or summer),
and thus has a semi-annual signal. Although the increase in the MOC to this end-of-year peak relative to the
magnitude of decrease from the preceding peak is smaller in the ensemble-mean than in SAMBA, it is noteworthy,

increasing by 52% of the preceding decrease (and by 77% in the seasonal cycle over 1993–2020) compared to 84% in SAMBA. The altimeter-based estimate has no significant increase in the geostrophic component in austral spring, and there is also no increase in the ensemble-mean MHT, unlike in SAMBA (Fig. 4, lower right panel).

### 3.3 Baroclinic and barotropic components

We investigate possible causes of the difference in variability between SAMBA and the ensemble by separating the geostrophic MOC anomalies into their baroclinic and barotropic components. The baroclinic and barotropic components of the MOC are not directly comparable between the ensemble and SAMBA due to differences in the reference level depth, but this probably has little impact on the differences between these estimates (see Section 2.2). The seasonal cycles of these components largely oppose each other in the ensemble with their sum equal to the geostrophic component (Fig. 5). In contrast, these components tend to augment each other in SAMBA (Fig. 5), so their geostrophic seasonal cycle has variations of a greater magnitude. The baroclinic component tends to dominate in both datasets, primarily controlling the phase of the geostrophic MOC seasonal cycle (Fig. 5). Although the barotropic component tends to oppose the baroclinic component in the ensemble, it has a notable effect on the phase of the geostrophic MOC seasonal cycle over 2013–2017, unlike over 1993–2020. Thus, while differences in the seasonality of the baroclinic MOC component account for most of the difference in the seasonality of the geostrophic MOC, differences in the barotropic component between the ensemble and SAMBA also play a role.

We also analyse the monthly-mean and inter-annual variations in the baroclinic and barotropic components of the MOC anomalies (Fig. 6). Both the baroclinic and the barotropic components of the MOC have similar monthly-mean variability in the ensemble and in SAMBA over 2013–2017 (Fig. 6d,e), although the baroclinic variability is slightly higher in SAMBA (7.5 Sv vs 5.3 Sv). Similarly, the inter-annual variability of the baroclinic and barotropic components has similar peak-to-trough magnitudes over 2013–2017 in the ensemble and SAMBA (Fig. 6f). However, since the barotropic component opposes the baroclinic component in the ensemble, the geostrophic and total MOC anomalies in the ensemble have much smaller monthly-mean and inter-annual variability than in SAMBA (Fig. 6a,b,f and Table 2). The monthly-mean and 12-month running mean baroclinic and barotropic components in the ensemble have even larger variability over 1993–2020, but these components oppose each other over the whole period (Fig. 6f).

### 4 Discussion

Seasonal variations in the baroclinic component of the MOC in the ensemble over 1993–2020 are caused by seasonal variations in both the eastern and western boundary volume transports, with variations in the western boundary tending to dominate. Over 2013–2017, there is much larger spatial variability in the seasonal transport, with significant contributions to the seasonal variations from the interior as well as from the boundaries. Therefore, differences in the MOC seasonality between datasets is likely caused by seasonal variations in both the boundary currents and the interior baroclinic transports. A spatial analysis of the baroclinic transports in SAMBA could determine the regions responsible for seasonality of this component and thus why it differs from the ensemble.

The altimeter-based estimate uses reference velocities at 1000 m depth that are constant in time. Thus, the barotropic component has no temporal variability so the geostrophic MOC anomalies only account for baroclinic

transport anomalies above 1000 m. Given the baroclinic component primarily determines the shape of the seasonal cycle in the ensemble and SAMBA, the fact the barotropic component is constant in the altimeter-based estimate may not significantly impact its estimate of the MOC's seasonal cycle phase. However, the magnitude of its monthly, inter-annual and seasonal variability may be affected if temporal changes in the barotropic component are important as suggested by the ensemble and SAMBA estimates of this component. The reference level depth used in the reanalyses (i.e., not in our baroclinic and barotropic component estimates, but that implemented in the models and thus in the geostrophic estimate) is the ocean floor, closer to the depths used to estimate the time-varying barotropic component in SAMBA. Thus, differences in the reference level are unlikely to cause the differences in the geostrophic component between SAMBA and the ensemble. However, differences in the methods used to estimate the barotropic velocity at that reference level could cause some of the difference.

We have shown that the monthly-mean MOC variability (i.e., standard deviation) is greater in SAMBA than in the ensemble and altimeter-based estimate, primarily because the Ekman, barotropic and baroclinic components augment each other in SAMBA, whereas they tend to be more opposed in the ensemble and altimeter-based estimates. While the standard deviation provides an insight into the month-to-month fluctuations, it does not determine the frequency of these fluctuations. Both the baroclinic and barotropic components have more frequent monthly fluctuations in SAMBA than in the ensemble (Fig. 5). These high-frequency variations could be caused by ocean eddy variability and variations that were previously under-resolved with only two mooring sites, and are now better resolved but likely still aliased with nine sites.

**5 Conclusions**

An ensemble of global ocean reanalyses from CMEMS provides a useful estimate of the magnitude and variability of the South Atlantic MOC and MHT, although it differs substantially from estimates based on SAMBA array data at 34.5°S, observed between 2013 and 2017. The ensemble is compared with several other estimates of the MOC and MHT, which differ in many aspects from, but also have similarities with, the reanalyses.

The ensemble-mean (and 1/12 degree GLORYS12V1 reanalysis) transports have no long-term trend over 1993–2020, although the trends in the individual reanalyses differ, and observational estimates increase over this period. All estimates of the time-mean MOC are similar (~15.5–18.7 Sv), but relative to the ensemble-mean value there is greater spread in the MHT (0.31–0.61 PW), with the ensemble-mean weaker than SAMBA observations. Monthly-mean MOC and MHT in the ensemble, the 1/12 degree GLORYS12V1 reanalysis, and an altimeter-based estimate (Dong et al., 2021) vary significantly less than those from the SAMBA array. In contrast, energy-budget estimates of MHT (Mayer et al., 2022) have a large monthly-mean variability comparable to SAMBA. Both the monthly-mean MOC and MHT in the ensemble are significantly correlated with the altimeter-based estimate across the whole 1993–2020 period (although most of the skill is from the seasonal cycle), whereas correlations with SAMBA estimates are not significant.

While there is inter-annual variability in the reanalyses and altimeter-based estimate over 1993–2020, SAMBA observations and some energy-budget MHT estimates have much larger inter-annual variability. The climatological seasonal cycles of the MOC and MHT vary considerably in phase and magnitude between estimates

due to differences in the geostrophic flow, with good agreement in the Ekman contributions among all datasets considered. Differences in the baroclinic component of the MOC are most important for determining the phase of the seasonal cycle in both the reanalyses and SAMBA, although the barotropic component also plays a role. The baroclinic, barotropic and Ekman MOC components tend to augment each other in SAMBA, whereas they tend to oppose each other in the ensemble and altimeter-based estimate. Thus, in SAMBA the monthly-mean, inter-annual and seasonal MOC anomalies have a greater magnitude than in the ensemble and altimeter-based estimate. This causes a large increase in the monthly-mean standard deviation of the total MOC in SAMBA. The baroclinic and barotropic MOC anomalies also have more frequent monthly-mean fluctuations in SAMBA.

Further insight into the cause of the similarities and differences between the ensemble, SAMBA and the altimeter-based estimate might be found by comparing the monthly-mean density profiles of these estimates. This could show how contributions by the baroclinic velocity to the geostrophic MOC anomalies vary between the datasets, including their spatial variations and how these lead to differences in seasonality. Similarly, the barotropic velocity (vertically averaged velocity) in the reanalyses can be compared with that used by the in-situ-altimetry and SAMBA methods to reference the flow. We also suggest exploring the horizontal resolution of SAMBA moorings used on the boundaries since it may alias variability here, with too few sites over steeply sloping topography. The impact of array resolution on SAMBA could be inferred by recalculating the baroclinic and barotropic components of the MOC in the ensemble using only a subset of their vertical density profiles. Reanalyses could therefore inform whether modifications to the observational density across the SAMBA array may provide more robust observational transport estimates. Use of the expanded set of moorings will also allow us to determine the importance of aliasing of variability on the boundaries. Since the reanalyses are in reasonable agreement with altimeter-based estimates but not with SAMBA, it prompts closer inspection of the methodologies used to make the computations.

To summarise, an ensemble of ocean reanalyses appears to be a useful tool to understand changes in the South Atlantic MOC and MHT, and to identify differences between observational estimates. Reanalyses also enable examination of variations prior to the SAMBA array record. Comparisons of reanalyses and observational estimates can be used together to refine methodologies and sampling approaches, and ultimately improve our understanding and estimations of ocean transports in the South Atlantic.

**Data Availability**

All data products used in this paper are listed in Table 1, along with their corresponding documentation and online availability.

**Competing Interests**

The authors declare that they have no conflict of interest.

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

| Ref. No. | Product name & type | Documentation |
|---|---|---|
| 1 | GLOBAL_REANALYSIS_PHY_001_031, Reanalysis (C-GLORSv7 and GLORYS2V4 ocean reanalyses) [1993–2020] | QUID: http://marine.copernicus.eu/documents/QUID/CMEMS-GLO-QUID-001-031.pdf<br><br>PUM: http://marine.copernicus.eu/documents/PUM/CMEMS-GLO-PUM-001-031.pdf |
| 2 | ORAP6 global ocean reanalysis [1993–2020] | Updated version of the ORAS5 reanalysis from GLOBAL_REANALYSIS_PHY_001_031 listed in Product Ref 1<br><br>The updated CMEMS global ocean reanalysis ensemble containing this reanalysis will be available online soon.<br><br>See Zuo et al (2021): https://doi.org/10.5194/egusphere-egu21-9997 |
| 3 | GloRanV14 global ocean reanalysis [1993–2021] | Updated version of the FOAM/GLOSEA5 reanalysis from GLOBAL_REANALYSIS_PHY_001_031 listed in Product Ref 1<br><br>The updated CMEMS global ocean reanalysis ensemble containing this reanalysis will be available online soon. |
| 4 | GLOBAL_MULTIYEAR_PHY_001_030 (GLORYS12V1 ocean reanalysis) [1993–2019] | QUID: https://catalogue.marine.copernicus.eu/documents/QUID/CMEMS-GLO-QUID-001-030.pdf<br><br>PUM: https://catalogue.marine.copernicus.eu/documents/PUM/CMEMS-GLO-PUM-001-030.pdf |
| 5 | South Atlantic Meridional Overturning Circulation – Basin-wide Array (SAMBA) observations for 2013–2017 (Kersalé et al. 2020 (for MOC), Kersalé et al. 2021 (for MHT) | https://www.aoml.noaa.gov/phod/research/moc/samoc/sam/data_access.php<br><br>https://www.aoml.noaa.gov/phod/SAMOC_international/samoc_data.php |
| 6 | Blended in situ and satellite altimeter estimates for 1993–2021 (Dong et al., 2021) | See Dong et al (2021): https://doi.org/10.1029/2020JC017073 |
| 7 | Energy-budget estimates of Mayer et al., 2022 [1993–2017] | Atmospheric energy budgets using ERA5 available at https://cds.climate.copernicus.eu/cdsapp#!/dataset/derived-reanalysis-energy-moisture-budget?tab=overview<br>TOA radiation data from University of Reading:<br><br>https://researchdata.reading.ac.uk/271/ |
| 8 | Energy-budget estimates of Trenberth et al., 2019 [2000–2016] | https://gdex.ucar.edu/dataset/Ocean_MHT_Values.html |

**Table 1: Data products used in this study, including documentation where available.**

| Variable | Statistic | Ocean reanalyses | | SAMOC Estimates | | Energy-budget estimates | | |
|---|---|---|---|---|---|---|---|---|
| | | Ensemble | GLORYS12V1 | SAMBA | Altimeter Dong | Trenberth | Mayer ORAS5 | Mayer ORAP6 |
| MOC (Sv) | Mean ± uncertainty (2013–17) | 16.56 ± 0.37 (16.29) | 18.72 (18.02) | 17.29 ± 5.0 | 18.69 | - | - | - |
| | Monthly-mean variability | 2.67 (3.20) | 2.90 (2.70) | 11.35 | 3.25 | - | - | - |
| | Mean ± uncertainty (1993–2020) | 16.38 ± 0.66 (16.11) | 19.23 (18.51) | - | 18.34 | - | - | - |
| | Monthly-mean variability | 3.00 (3.53) | 3.30 (3.14) | - | 3.48 | - | - | - |
| | Trends (Sv/decade) (1993–2020) | 0.17 (NS) | -0.08 (NS) | - | 0.66 | - | - | - |
| MHT (PW) | Mean ± uncertainty (2013–17) | 0.36 ± 0.03 | 0.44 | 0.50 ± 0.23 | 0.61 | - | 0.31 | 0.31 |
| | Monthly-mean variability | 0.19 | 0.20 | 0.55 | 0.20 | - | 0.46 | 0.43 |
| | Mean ± uncertainty (1993–2020) | 0.37 ± 0.04 | 0.49 | - | 0.58 | 0.33 (2000-16) | 0.33 (1993–2017) | 0.34 (1993–2017) |
| | Monthly-mean variability | 0.20 | 0.23 | - | 0.21 | - | 0.40 | 0.44 |
| | Trends (PW/decade) (1993–2020) | -0.001 (NS) | -0.007 (NS) | - | 0.036 | -0.010 (2000-16) | 0.086 (1993–2016) | 0.094 (1993–2016) |

| Variable | Statistic | Ensemble | | | |
|---|---|---|---|---|---|
| | | GloRanV14 | C-GLORSv7 | ORAP6 | GLORYS2V4 |
| MOC (Sv) | Trends (Sv/decade) (1993–2020) | 1.18 | -0.32 | 0.41 | -0.60 |
| MHT (PW) | Trends (PW/decade) (1993–2020) | 0.042 | -0.014 (NS) | -0.012 (NS) | -0.016 (NS) |

**Table 2: Time-mean and uncertainty (or ensemble spread), monthly-mean variability and trends of the maximum MOC and the MHT across 34.5°S, for the ensemble-mean (product ref's 1, 2, 3),**
**GLORYS12V1 (product ref 4), SAMBA observations (product ref. 5), an altimeter-based estimate (product ref 6) and energy-budget estimates (product ref's 7 and 8). All volume transports are referenced to zero at the surface. The time-mean MOC and monthly-mean variability calculated in the reanalyses using no net zero transport constraint is added in parentheses. Time-mean values are calculated over the 2013–2017 SAMBA observational period and over the full 1993–2020 ensemble period, if available.**
**Uncertainty in the ensemble-mean is defined as the standard error of the time-mean transport across the ensemble (note: this is smaller than the true uncertainty in the estimate). Monthly-mean variability (i.e., a measure of the deviation of monthly-mean data from the time-mean) is defined as the standard deviation of the monthly-mean transports over the timeseries. Methods used to calculate SAMBA observational uncertainty** (Kersalé et al., 2021, 2020) **are described in** Meinen et al., 2013 and Kersalé et al., 2021. **Trends that are statistically insignificant (p>0.05) are labelled NS.**

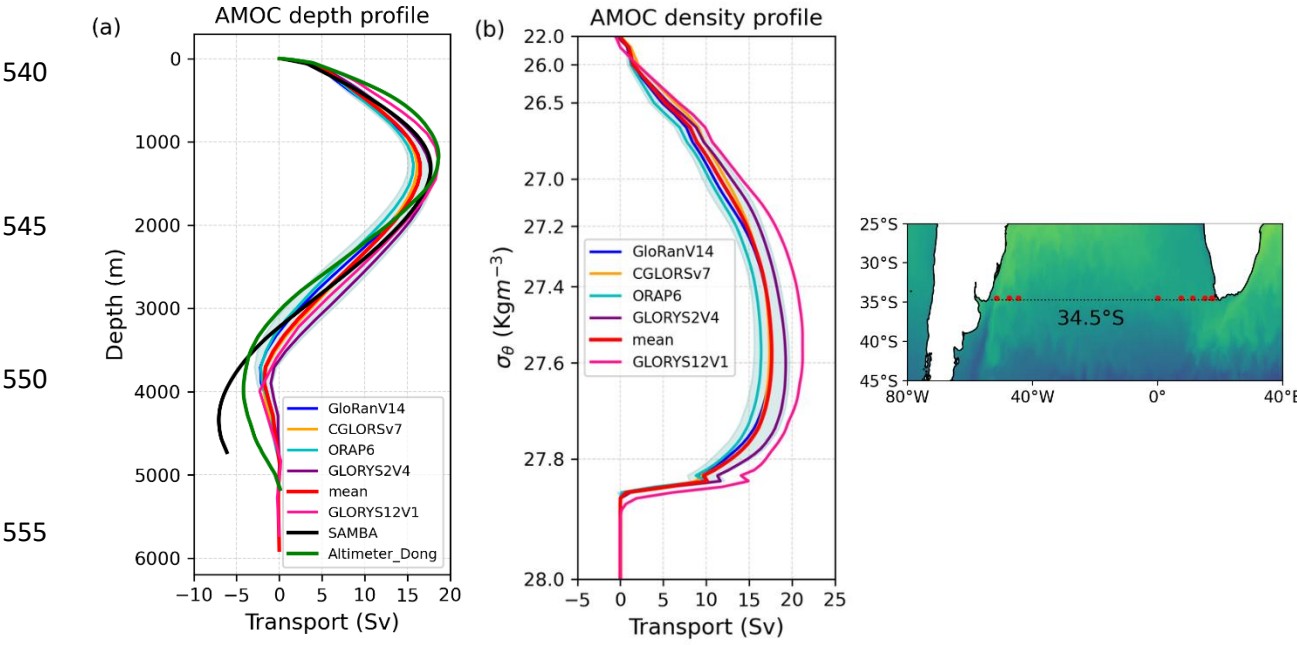

**Figure 1: Vertical profile of the overturning transport across 34.5°S in (a) depth space and (b) density space, averaged over the 2013–2017 period of SAMBA observations, from September 2013 to July 2017. The reanalyses ensemble-mean (red, product ref's 1, 2, 3) and spread (light cyan shading) are plotted, along with each ensemble member, the GLORYS12V1 reanalysis (pink, product ref 4), the SAMBA estimate of** Kersalé et al., 2020 **(black, product ref. 5) and an altimeter-based estimate of** Dong et al., 2021 **(green, product ref 6). The ensemble spread is defined as two times the standard deviation across the ensemble members. (right panel) Map showing the location of the SAMBA moorings (red dots) along 34.5°S.**

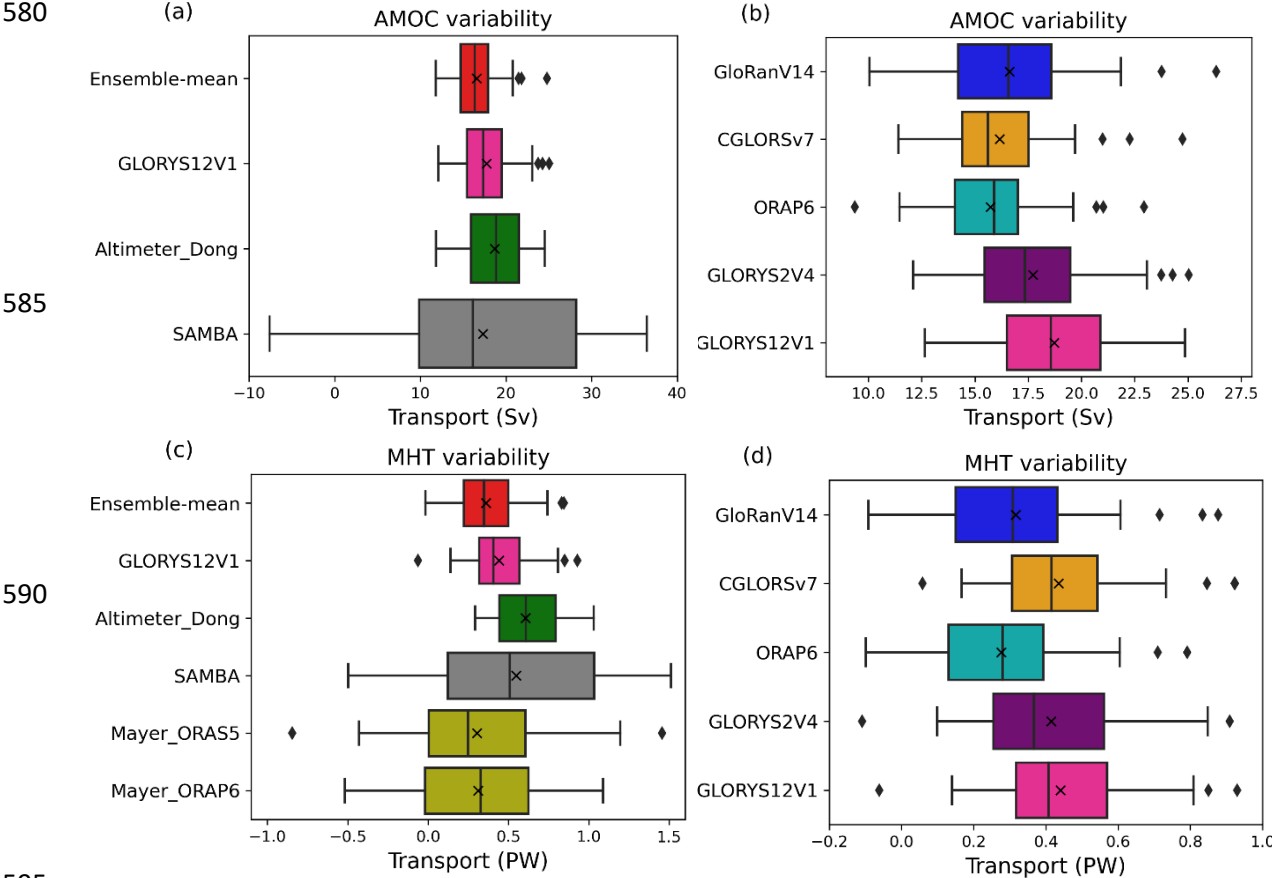

**Figure 2: Whisker-box plots of the monthly-mean MOC (top panels) and MHT (bottom panels) across 34.5°S, over the SAMBA observational period (2013–2017), using the same products as in Fig. 1. Energy-budget estimates, "Mayer_ORAP6" and "Mayer_ORAS5", (yellow, product ref 7) are also used for the MHT. Reanalyses analysed are shown in (b) and (d) with a reduced scale to highlight the differences between models. Boxes represent the interquartile range (IQR) with the median (line) and mean (crosses) shown. Whiskers cover a range of values up to one IQR beyond the upper and lower quartiles, and diamonds are outlying values beyond this range. Note: the x-axis scale changes between the left- and right-hand plots.**

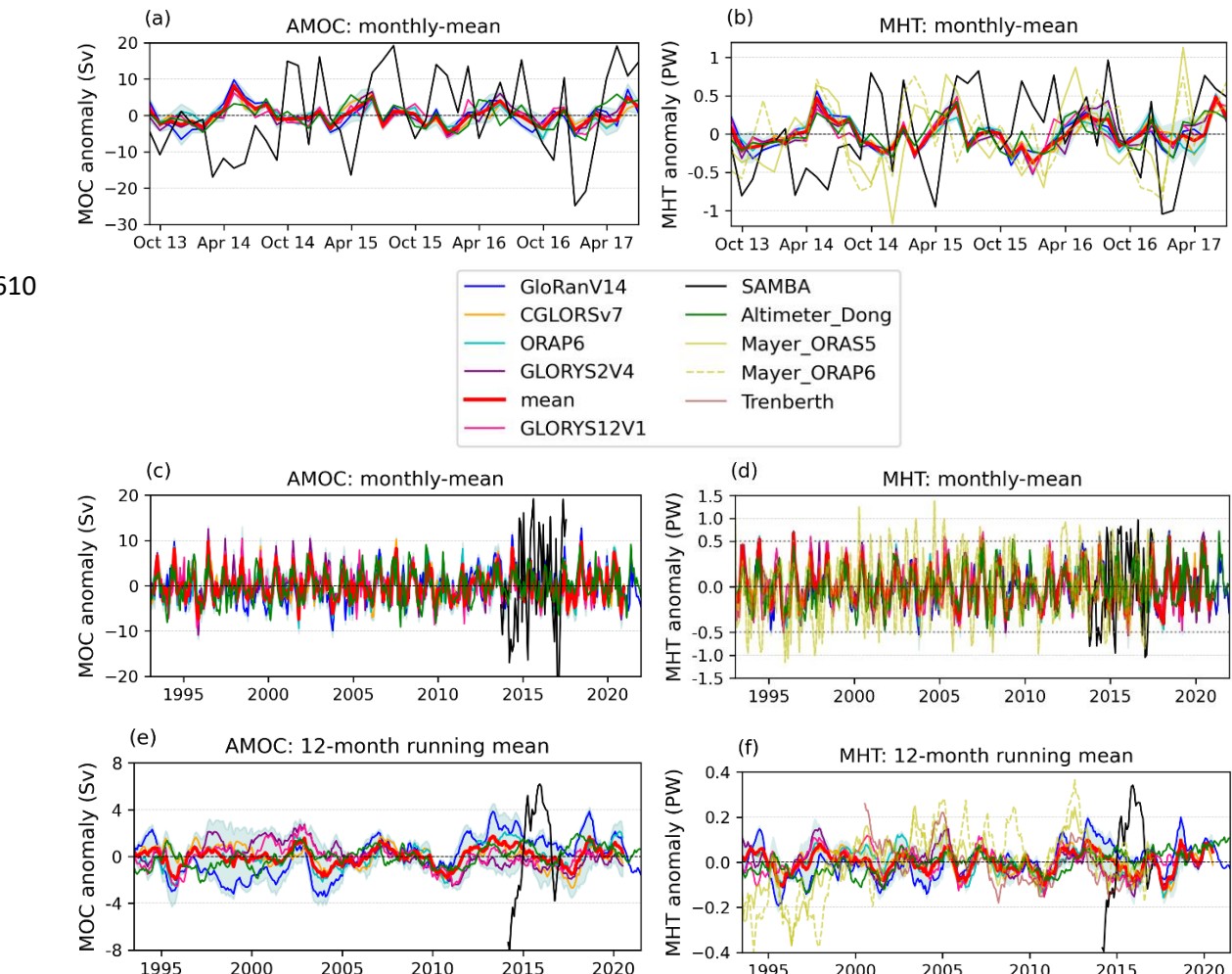

**Figure 3: Timeseries of the monthly overturning (left) and heat transport (right) anomalies nominally across 34.5°S, with monthly-mean values from September 2013 to July 2017 (top panels) and over 1993–2021 (middle panels), and 12-month running mean values over 1993–2021 (bottom panels), in the four reanalyses, ensemble-mean (red), GLORYS12V1 (pink), SAMBA observations (black), an altimeter-based estimate (green) and energy-budget estimates (yellow and brown, product ref 8). Labels, shading and product information as in Fig. 1. The horizontal grey dotted lines in (d) divide the y-axis into two linear scales, with the y-axis compressed above the line. Note: Trenberth energy-budget estimate is for latitude, 33.5°S.**

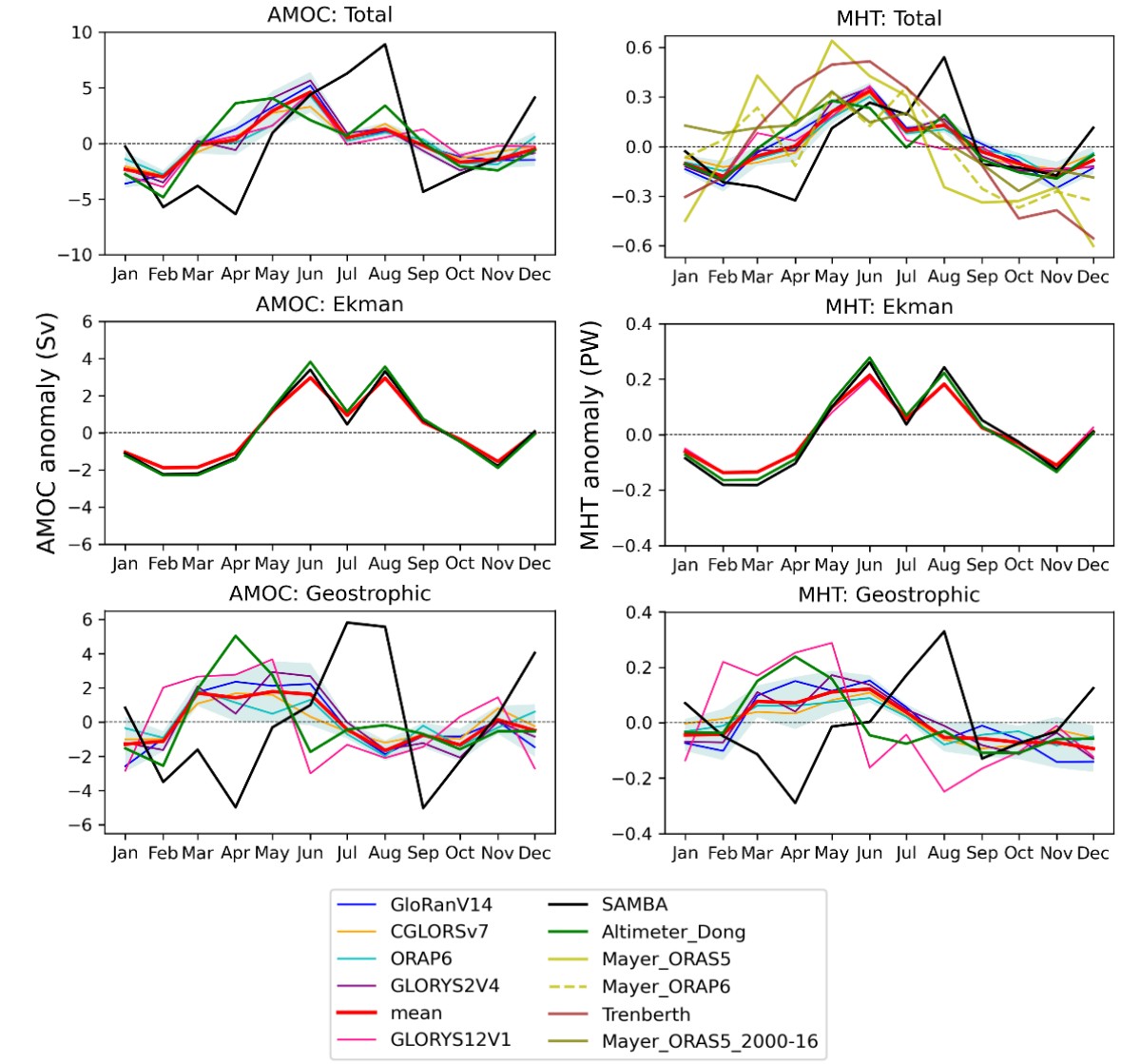

**Figure 4: Seasonal cycles of (left) the overturning and (right) the MHT anomalies across 34.5°S, averaged over the SAMBA observational period from September 2013 to July 2017. The exception is the energy-budget MHT estimate of Trenberth et al., 2019, which is averaged over 2000–2016, and also the ORAS5-based Mayer energy-budget estimate, "Mayer_ORAS5_2000–16" (olive), is averaged over the same period for comparison. The total (top panels), Ekman (middle panels) and geostrophic (bottom panels) components of these transports are plotted. Labels, shading and product information are as in Figs 1 and 3.**

**2013-2017**            **1993-2020**

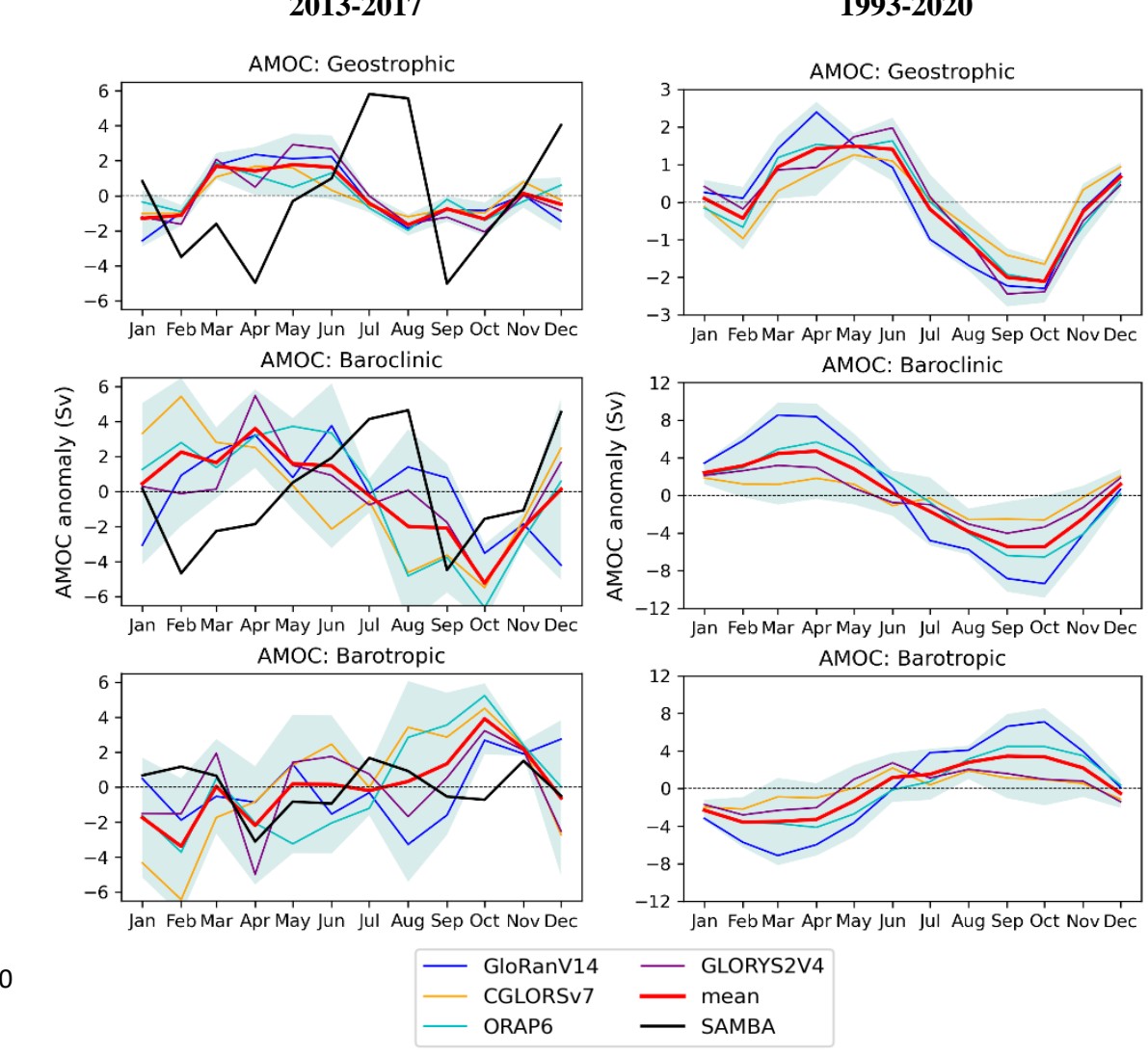

**Figure 5: Seasonal cycles of the MOC anomalies across 34.5°S, averaged over the (left) the SAMBA observational period from September 2013 to July 2017 and (right) the 1993-2020 period of the reanalyses. The geostrophic (top panels), baroclinic (middle panels) and barotropic (bottom panels)**
**components of these transports are plotted. Labels, shading and product information are as in Figs 1 and 3.**

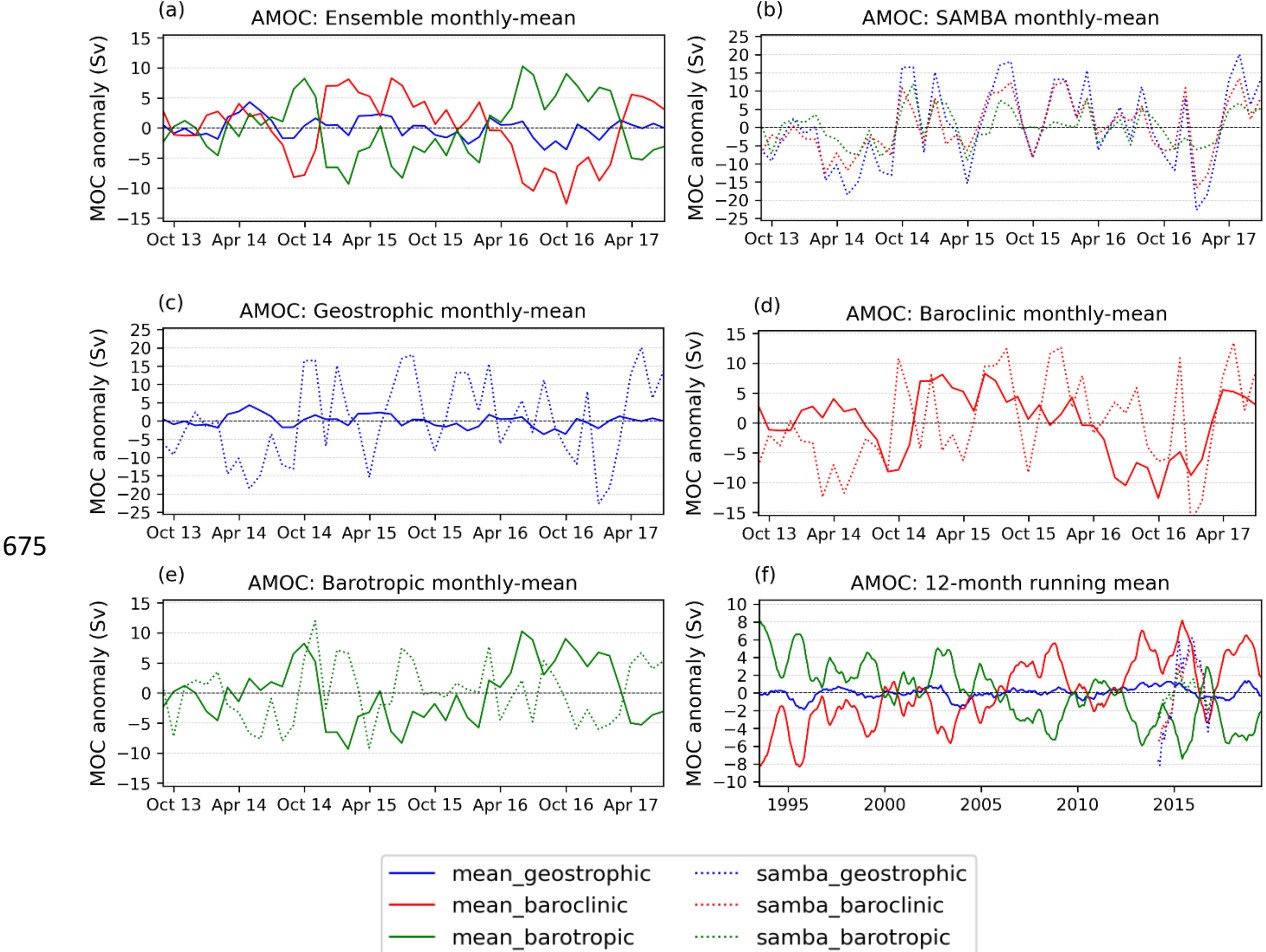

**Figure 6: Timeseries of (a) - (e) Monthly-mean (2013-2017) and (f) 12-month running mean (1993-2019) MOC anomalies showing the geostrophic (blue), baroclinic (red) and barotropic (green) components in (a) the ensemble, (b) SAMBA and (c) - (f) both the ensemble (solid lines) and SAMBA (dashed lines).**