# Peer review of "South Atlantic overturning and heat transport variations in an ocean reanalysis ensemble and observation-based estimates"

_State of the Planet, 2022_

## Referee Comment (RC2)

This manuscript describes the results of study that is a comparison of an ensemble of model re-analysis and recent observation based (XBT, mooring, satellite SSH) estimates of meridional overturn and heat transport across 34.5°S, the location of the SAMBA array. The paper is well organized and well-written and is worthy of publication. My main concern is that prior to the discussion of the seasonal and interannual cycles, it is simply a comparison, with little to no investigation into causes of similarities or differences. I believe there is the potential for greater impact with some more detailed discussion and/or reasoning in the earlier part of Section 3 or perhaps this section could be broken up into two sections "presentation of results" and "discussion of results".

Please find my comments below. Note they are written in the order in which they came up, which may indicate to the authors that certain topics should be mentioned sooner rather than later.

Line 65: Whose AMOC estimates? To what does this refer?

Line 66-77: I think the author's ought to be more careful about lumping all observation based estimates into two simple categories – satellites and in-situ, because some of the latter are based on observations of transport, some on observations of T&S, some on T with other assumptions to get at geostrophy, and some of completely different assumptions to estimate MOC/MHT from floats.

I realize it is a recent publication and does not include a line at 34.5°S, but I would recommend including a reference to Cainzos et al (2022) in this paragraph as it represents a recent investigation using another prevalent means of estimating the MOC and MHT from in situ observations. https://agupubs.onlinelibrary.wiley.com/doi/full/10.1029/2021GL096527 Authors' choice of course.

Line 103: Is there a way to consider how biased all these reanalysis products are to surface observations. That is, how important are subsurface observations to their estimates? I ask because it concerns me that the near surface reanalysis product may be much improved compared to the model alone, but can the same be said for the deep ocean?

Line 124: Wouldn't one expect a net southward transport in the Atlantic? I would have thought the net zero constraint should be applied globally.

Line 127-128: "Each observational product applies its own constraint to reference the flow." This may be true, but I don't understand the intention of this statement here.

Figure 1: Where is AABW?

Line 144: …due to the existence of fewer observations …  (is this discussed elsewhere)

Line 146-147: density space versus pressure space. I completely agree, but this is not new observation nor is it specific to 34.5°S. Take a look, for example, at Lumpkin and Speer (2007) or the many works of Susan Lozier.

Line 148-149 – It seems this is an important point. Is there no way to include an illustration of this lack of sensitivity, even if it is shown as supplementary material?

Line 155: Perhaps I am misunderstanding. Given the overlap, I don't see how one can make any comparison at all amongst these products. I also don't understand how one can talk about an "increased spread in the time-mean MHT compared to the MOC." What scale is being used?

Line 170: Again, I feel like I am missing something here. Don't the ensemble means have the satellite record embedded in their estimates? If so, it would indicate a major issue if the two were not strongly related. Okay, I see the discussion Lines 180-184, but I would still argue that reanalyses are so dependent of the satellite record and that their characteristics must by necessity be closer to the near-surface focused observations. If this is the case, perhaps lines 180-184 should be the lead-in to the discussion.

Lines 192-195 – Perhaps there are too many curves or the figure is too squashed, I cannot see these trends in Figure 3. The same is true for the trends noted in the next paragraph.

Lines 205-210 – which begs the questions are the models and satellite estimates getting the boundary currents and is there something important in the middle of the basin that SAMBA is missing?

Line 274-279 – These sentences come off as an excuse not provide something thinking and analysis into the causes of differences or similarities. The true impact will come with some attempt to answer these questions.

Trivia:
Line 190: … running mean overturn and MHT … **are** relatively…
 And elsewhere - "overturn" is the noun, "overturning" is the adjective

Table 1, Figures 1 & 2 and text:  The SAMBA and Dong estimates. It took me a while to find the statement (lines 104-106 stating whose estimates were being used.  These references should also be cited in at least one of the figure captions and in the succeeding captions the reader should be referred back to the figure that includes the citations.

---

## Author Comment (AC1)

**Response to reviewer comments for manuscript: "Overturning and heat transport variations in the South Atlantic in an ocean reanalysis ensemble and other estimates"**

We thank both reviewers for the constructive feedback and ideas of where we can improve the study. We note that the Ocean State Report articles are limited to 3600 words (main text) and four figures, though we are able to add Supplementary material, two additional figures and expand the manuscript by up to 20% at this stage. We have expanded the discussion of the cause of differences between the datasets by analysing the baroclinic and barotropic components of the MOC in the ensemble and comparing this with the SAMBA observations. We also provide ideas of future work that could further determine the differences, for example, by comparing the vertical density profiles and by exploring the horizontal resolution of SAMBA moorings used on the steeply sloping western and eastern boundaries. We believe the further analysis supports our conclusion that reanalyses appear to be useful to understand changes in the MOC and MHT and differences between the datasets. We also note that the fact the reanalyses is in good agreement with the variability of the altimeter-based estimates suggests they are useful, since the altimeter-based estimates are one of the main observational datasets that are used to assess the state of the MOC and MHT in the South Atlantic. Moreover, because these reanalyses agree with one set of observations (altimeter-based estimates) vs. another set of observations (SAMBA), it prompts closer inspection of the details of the methodologies used to make those computations which can point to areas where the barotropic and/or baroclinic variability may be aliased or under-sampled. In the absence of agreement between the reanalyses and one of the observation-based estimates, it was less clear how to begin assessing the discrepancy between the observational approaches.

**Reviewer 1**

The manuscript presents a synthesis of the performance of an ocean reanalysis ensemble concerning the strength and variability in the meridional overturning circulation (MOC) and the meridional heat transport (MHT) across 34.5S in the South Atlantic. It includes several observation-based estimates for comparisons that reveal large model-observation and model-model discrepancies. The main conclusion is that a high level of uncertainty remains in the South Atlantic MOC and MHT from the ocean reanalysis products. The results will help understand the model's efficacy in capturing the large-scale circulation variability in the South Atlantic. They will likely provide important feedback to the MOC observing system in the region. I recommend it be accepted for publication after the following issues have been addressed.

We thank the reviewer for their positive assessment of the manuscript and insightful comments. We respond to their comments and questions below in blue font.

Main comments:

The manuscript presents detailed comparisons for the MOC/MHT estimates (e.g., the mean and variability and the trend). However, it is not clear what one can learn from the agreement and disagreement between ocean reanalyses and observations. The presented

analysis may not support the main conclusion that "an ensemble of ocean reanalyses appears to be a useful tool to understand changes in the South Atlantic overturning and heat transport, and differences between estimates." In my opinion, the authors will need to include more discussion on the model-observation differences, and be careful and clear about what can be inferred from them, e.g., that will be useful in future analyses based on those ocean reanalyses.

The similarity between the reanalyses and the in situ-altimeter estimate vs. the SAMBA nine-mooring estimate suggests the reanalyses could be used to determine why the two observational estimates are different. It is unclear whether SAMBA nine-site is over-estimating/aliasing the spatial and temporal variability, or the in situ-altimeter estimate is under-estimating/aliasing the variability, or both. We have a hint that both are true, because the SAMBA estimate (Meinen et al., 2018) using the two most inshore SAMBA mooring sites has a lower standard deviation than the nine-site estimate (Kersalé et al., 2020, 2021), but still has a larger standard deviation than the in situ-altimeter estimate.

Reanalyses are useful because we have complete temperature, salinity, and velocity fields in 3D space (no spatial aliasing), a known barotropic velocity, and reanalyses are constrained by some of the same observations (altimetry, temperature, and salinity) that are used to produce the in situ-altimeter estimates. However, reanalyses may still underestimate some of the higher frequency/small-spatial-scale variability in the real ocean.

Still, we can explore and test hypotheses about why the reanalyses differ from the SAMBA estimates. In the model, the geostrophic transport can be separated into its baroclinic and barotropic components by using the vertical profiles of density and the geostrophic equations to obtain the baroclinic component and calculating the barotropic component as a residual using the model velocity. Thus, we can examine whether the differences are in the baroclinic velocity (i.e., density) structure or in the barotropic velocity used to reference the flow. We significantly revised portions of the manuscript to address as many of the above points as possible in the short space allotted (see Figure 5 and Figure 6 in the revised manuscript, and the discussion of those figures).

Although beyond the scope of this study, the vertical profiles of density (and temperature and salinity) in the reanalyses, SAMBA estimate, and altimeter-based estimate could be directly compared to determine how and if they differ. Also, estimates of the baroclinic component using different density profiles with different zonal spacing could be used in the reanalyses to infer the impact of the different resolution of the datasets. Analysis using the vertical density profiles may show the cause of differences in the long-term mean, monthly, and inter-annual variability and the seasonal cycles (over 2013-2017 and 1993-2020) between estimates. Similarly, the reanalyses velocity can be used to estimate the barotropic velocity (vertically averaged) along 34.5S, which can then be compared with the reference velocity used by the in situ-altimetry and SAMBA methods to reference the flow. Analysis of the two-dimensional fields of baroclinic and barotropic velocity along 34.5S would allow spatial differences in the velocity fields between estimates to be inferred, enabling further understanding of the cause of differences. Specifically, we could examine how they differ on the boundaries over sharply sloping topography and in the interior. We view this as an important next step to take in future studies. The need for delving into these details has

been heightened as a result of model-observational comparisons such as the one conducted in this study. We have added a paragraph to describe these ideas for future work based off reanalyses in the Conclusions section of the revised manuscript.

Minor comments:

Lines 119-130: How does this constraint affect the MOC and MHT estimates in ocean reanalyses? Does applying such a constraint alter the variability? over what time scales?

The net zero volume transport constraint has a small but significant impact on the MOC estimate based on a comparison of estimates from the reanalyses calculated with and without applying the net zero volume constraint. When the constraint is not applied, we integrate meridional velocity from the surface to the ocean floor, so the MOC estimate is consistent with the other datasets that assume the vertical profile has zero transport at the surface. The time-mean MOC and monthly-mean MOC variability (i.e., the standard deviation of monthly-mean values) with and without applying the constraint are as follows:

| | Time-mean MOC (Sv) | | Monthly-mean MOC Variability (Sv) | |
|---|---|---|---|---|
| | Constraint | No Constraint | Constraint | No Constraint |
| Ensemble (2013-2017) | 16.56 | 16.29 | 2.67 | 3.20 |
| Ensemble (1993-2020) | 16.38 | 16.11 | 3.00 | 3.53 |
| GLORYS12V1 (2013-2017) | 18.72 | 18.02 | 2.90 | 2.70 |
| GLORYS12V1 (1993-2020) | 19.23 | 18.51 | 3.30 | 3.14 |

These values are added in brackets to Table 2.

There are significant differences in both the time-mean and monthly-mean variability when the constraint is used. However, the changes in monthly-mean variability in the ensemble and GLORYS12V1 are small compared to differences between these estimates and SAMBA.

The correlation (p<0.05) between the timeseries calculated without the constraint and with the constraint over the full 1993-2020 period and over the SAMBA observational period is 0.96 and 0.97 respectively.

The net zero volume constraint must be applied for the MHT to allow for an estimate in the expected units of heat transport (i.e., Watts). This is consistent with SAMBA (which assumes the volume transport below ~4700 m balances the transport above in the MHT estimate) and the altimeter-based estimate. It also enables comparison with the energy-budget estimate.

Line 134: What is the typical Ekman depth at 34.5S? How to justify the use of SST instead of temperature over the Ekman layer?

We now state that we assume SST is representative of the temperature of the Ekman layer: "In the reanalyses, the Ekman component is calculated using the ERA5 or ERA-Interim wind stress, and for MHT, the zonal-mean SST across the section, assuming SST is representative of the Ekman layer temperature."

The Ekman transport is greatest near the surface and decreases towards the bottom of the Ekman layer. The Ekman layer is less than 50 m at 34.5°S, calculated from the equation, $D_{E\_} = sqrt(7.6/sin(phi)) \, U_{10,}$ assuming a typical wind speed at 34.5°S of 2 m/s based on ERA-Interim reanalysis (Lin & Munday, 2020). Thus, changes in temperature over this depth are expected to be small due to it being well mixed. We also note that since the Ekman transport decreases with depth, SST may be more appropriate than a depth-averaged temperature.
Dong et al. (2021) also use SST in their calculation of the Ekman component. In SAMBA, the Ekman transport is distributed equally over the upper 60 m of the water column with the temperature profiles reconstructed at the mooring sites and estimated between moorings using altimetry-hydrography relationships. Thus, all of these estimates have some error in this component, but since they are all similar in our analysis, this choice does not have a significant impact on our findings. Since we are focused on what happens throughout the ocean depth, the specific details in the Ekman layer are less important.

Lines 146-147: Please elaborate why "the overturning in density pace ... important to accurately infer the overturning pathways." And it is even more puzzling that if overturning in density space is preferable, how the authors decided to go with overturning in depth space. Please help clarify.

We have removed this statement because as pointed out by reviewer 2, this is not a new observation, and we do not think it is necessary to go into details on overturning pathways given this is not particularly relevant to the study.

The MOC calculated in density space is more accurate when determining the overturning pathways because water masses tend to flow along isopycnals through much of the ocean interior, with a largely adiabatic upwelling in the Southern Ocean, and transports by eddies are more accurately represented in density space. However, it is more important to use density space in subpolar regions where the isopycnals outcrop and are tilted vertically. In the subtropics, the isopycnals are flatter, so it is "less bad" to use depth coordinates in these regions.

We focus on the MOC calculated in depth space to allow for direct comparisons between the reanalyses and the other observational products that are only provided in depth space. We have expanded the following sentence to make this clear: "The temporal variability of their upper MOC strength at 34.5°S, however, is fairly insensitive to the vertical coordinate system used for integration (Fig. S1). We therefore focus on the MOC in depth space because the reanalyses can then be directly compared with the observational estimates."

We have added Fig S1 to the Supplementary Material to show the high correlation between the monthly-mean and 12-month running mean MOC in depth and density space at 34.5°S.

Line 156: How is the MHT range "in contrast" to the MOC range? They are different quantities; it is impossible to compare their ranges. Instead, my suggestion for the authors is to calculate and compare the MHT's dependence on the MOC at 34.5S between models and between model and observation.

We mean there is greater spread between the datasets in the time-mean MHT than in the time-mean MOC, relative to the ensemble mean value. The MHT has a 75% increase from the minimum to the maximum values (0.61 PW – 0.36 PW) relative the ensemble-mean of 0.36 PW, whereas the MOC has only a 20% increase (18.72 Sv – 16.56 Sv) relative to the ensemble-mean of 16.56 Sv. We agree this was confusing, so we now make clear this is relative to the ensemble-mean value. Even though they are different quantities, a 75% higher MHT would be considered large whereas a 20% increase in the AMOC is more reasonable.

We also compare the correlations between the MHT and MOC as suggested. We calculate the correlations between the monthly-mean values of MOC and MHT for each dataset. The correlations are high for all estimates, largest for SAMBA (r=0.96), whereas the in-situ and altimeter product and ensemble-mean have 0.91 and 0.90 (similar over the whole 1993-2020 period). We note these in lines 238-240.

Lines 187-189:  It is hard to understand how spatial resolution affects the long-term mean but not the monthly mean. Please elaborate.

This was unclear. We wrote that fully resolving eddies "has minimal impact on the amplitude and variation of the monthly-mean transports". This is confusing since the magnitude of the monthly-mean transports do differ between the eddy-permitting and eddy-resolving reanalyses causing the difference in the long-term mean. We were referring to the amplitude of the monthly-mean variability (i.e., the standard deviation). We have removed "amplitude" from this sentence, rewriting it as, "Thus, fully resolving (as opposed to only permitting) eddies in the ocean reanalyses considered here is important to infer the time-mean transports across 34.5°S, but has minimal impact on the variation of the monthly-mean transports."

Lines 197-201:  For a robust comparison, it would be better to calculate the MHT trend for the overlapping 2000-2016 period.

We agree this would be a more robust comparison of the reanalyses and altimeter-based estimate with the energy-budget estimates. We chose to show the 1993-2020 period trends since this includes more recent years. We therefore keep these estimates, but we have added an additional sentence stating the trends in GloRanV14 (the only reanalysis with a significant trend) and the altimeter-based estimate over 1993-2016: "The aforementioned MHT trends are similar over 1993–2016 (GloRanV14: ~0.047 PW/decade; altimeter: ~0.032 PW/decade)."

Lines 201-204:  It doesn't seem to belong to this paragraph.

We agree and have moved this sentence, below **line 183 in the original manuscript**, where we discuss the monthly-mean correlations.

Lines 207-208: How to understand the large differences between SAMBA and other estimates? Please refer to my main comments.

We agree this paper did not explain the differences between the estimates. We now compare the baroclinic and barotropic components in the ensemble with SAMBA to narrow down the cause of the differences (in Figure 5 and 6 and in the manuscript text). We discuss these differences in lines 262-314 showing that the phase and magnitude of the seasonal cycle of the baroclinic MOC component differs in SAMBA and the ensemble. The barotropic component also differs between the estimates, but the baroclinic component is most important in determining the seasonal cycle of each estimate.

Differences may also be due to the way that the baroclinic flow is referenced in the reanalyses (i.e., zero net mass transport) vs. in the observations (i.e., the use of a reference velocity). The observations take different approaches to referencing the flow and this may be yet another source of difference. This is an important subject for future analysis. Our analysis narrows down the possible causes of the difference – and could lead to modifications to the referencing of the flow in the SAMBA and in situ-altimetry estimates reconciling some of the differences between the estimates. It also shows that reanalysis are a useful tool to identify, characterize, and understand the differences, pointing both the observational and modelling communities towards the sources of the differences and the development of improved/consistent methodologies. We then suggest further work that could be done to determine the cause of the differences in lines 343-356.

We suggest that finer horizontal resolution of the datasets over the sloping topography of the continental shelves may be critical for resolving well the barotropic and baroclinic variations on the western and eastern boundaries. There was a drop in the number of locations for which data was being collected at SAMBA moorings from the middle of 2017 – 2021. The number of active moorings has increased since then, and this will allow us to examine where the nine-site SAMBA calculation may have aliased variability on the boundaries.

[Figure]

Figure: Timeline of SAMBA mooring deployments as of August 2022. Black solid (dashed) lines show record lengths of hourly (daily) data collected. Black circles show the date of the most recent deployment (not including cruise happening western boundary cruise in December 2022 as these observations are still being quality controlled).  In October 2023, when data from the western, interior, and eastern boundary moorings have been collected there will possibly be two-years of overlapping data from eight moorings on the west, three moorings in the interior, ten moorings on the east. This will allow us to compare different configurations of the array. The Kersale et al. (2020, 2021) SAMBA MOC/MHT estimates shown in this paper used data from A, C, D, P8/S, P6/U, P5/V, P4/W, P2/Y, P1/Z. Meinen et al. (2018) used data from A and P1/Z, which allows for the generation of a longer data set.

Lines 219-221:   It would be good to perform the significance test on the seasonality (e.g., using ANOVA).

Data collection from the western side of the SAMBA array has been hampered by cruise cancellations during 2020-2022 (see figure above) at key sites needed to update the MOC calculation. With an additional five-years of data collected since the estimates of Kersale et al. (2021), we can consider using techniques such as the one suggested by the reviewer to examine the sensitivity of the seasonal cycle to the number of years of data used.

Line 228:  "interannual variations in the seasonal cycle…"?  Please rephrase.

We have rewritten this sentence as: "Year-to-year variations in the annual cycles of each estimate over 2013–2017 (not shown), and differences in the climatological seasonal cycle between each estimate (Fig. 4)…."

Line 231:  Is the difference in the geostrophic component caused by the different reference levels?  More discussion will be needed on what we can learn from this.

We do not believe differences in the depth of the reference level are responsible for the differences between the ensemble and SAMBA (as noted in lines 294-305), but the method of estimating the flow at that reference level may be the source of some of the differences. Since we calculate the geostrophic component in the reanalyses as a residual of the total transport minus the Ekman transport, the reference levels are not used, and the geostrophic component will contain both the baroclinic and barotropic anomalies. For SAMBA, the barotropic component variability is calculated using the bottom pressure differences between profiles measured by two adjacent pressure inverted echo sounders and is applied at the depth of the shallowest of the two moorings. The time-mean barotropic component is taken from a model and is applied at a fixed reference level. Although the choice of reference level where the model mean flow is applied in their method may cause differences in the time-mean estimate, it will not affect the variability. However, the way the flow is referenced using bottom pressure gradients may explain some of differences in the geostrophic MOC anomalies between the ensemble, SAMBA, and the in-situ-altimetry method.

The Dong et al. (2021) altimeter-based estimate uses reference velocities at 1000 m depth that are constant in time. Thus, the barotropic component has no temporal variability. This may explain some of the differences between the altimeter-based estimate and SAMBA

since the altimeter-based geostrophic component only accounts for the baroclinic component.

Our analysis suggests the baroclinic anomalies determine the shape of the seasonal cycle in both the ensemble and SAMBA. Differences in the shape of the seasonal cycle also have contributions from the barotropic anomalies, likely due to differences in the barotropic reference velocity component, but not due to differences in depth of the reference levels.

We believe the distance between adjacent density profiles used for SAMBA is likely important, especially over rapidly sloping topography where there can be large bottom triangles over which the flow is not observed/resolved. This may introduce baroclinic geostrophic velocity variations over the continental slopes that would cancel out if more sites were used, but do not because only a couple of moorings span the region where the depth of the topography jumps from ~1000 m to ~4500m.

We now add a discussion of the reference level and reference level methodology to the paper in lines 144-156 and 294-305.

Lines 281-284: Those statements don't seem accurate and need rephrasing. Please refer to my main comments.

We believe this statement is now supported by our additional analysis of the baroclinic and barotropic components and our suggestion of future work to understand these differences through the reanalysis data e.g., analysis of the impact of different spatial differences in the SAMBA array by using data interpolated to the SAMBA array points in the reanalysis and by comparing their vertical density profiles. The ensemble also provides comparable results to altimeter-based data that has been used in many studies to understand the South Atlantic transports, which further supports the usefulness of the reanalysis estimates

Moreover, because these reanalyses agree with one set of observations vs. another set of observations, it prompts closer inspection of the details of the methodologies used to make those computations which can point to areas where the barotropic and/or baroclinic variability may be aliased or under-sampled. In the absence of such observations, it was less clear how to begin assessing and interpreting the discrepancy between the observational approaches.

We have rewritten this paragraph in lines 357-361.

RC2: ['Comment on sp-2022-8'](), Anonymous Referee #2, 13 Nov 2022   reply
Note, I have chosen major revisions only because I am not sure of editor's distinction between major and minor. I do think there needs to more discussion of possible causes behind the results presented in the early part of Section 3. See the attached file.

This manuscript describes the results of study that is a comparison of an ensemble of model reanalysis and recent observation based (XBT, mooring, satellite SSH) estimates of meridional overturn and heat transport across 34.5°S, the location of the SAMBA array. The paper is well organized and well-written and is worthy of publication. My main concern is that prior to the discussion of the seasonal and interannual cycles, it is simply a comparison, with little to no investigation into causes of similarities or differences. I believe there is the potential for greater impact with some more detailed discussion and/or reasoning in the earlier part of Section 3 or perhaps this section could be broken up into two sections "presentation of results" and "discussion of results".

We thank the reviewer for their positive assessment of the manuscript and insightful comments. We respond to your comments and questions below in blue font. We address your main concern in the revised version of the manuscript, where we delve into some (but not all given the limited length of the manuscript) of the causes of the similarities or differences between the observations. We also address your main concern in the responses to your individual comments below.

Please find my comments below. Note they are written in the order in which they came up, which may indicate to the authors that certain topics should be mentioned sooner rather than later.

Line 65: Whose AMOC estimates? To what does this refer?

The AMOC estimates from SAMBA over 2013-2017 that we use in this study were calculated by Kersale et al. (2020). However, volume transports have continued to be measured across SAMBA since 2017, so new data will be released at some point and future studies will use this data to calculate the AMOC. We have rewritten this sentence to make it clear that SAMBA data measured after 2017 have not yet been used to estimate the AMOC:

"The SAMBA array has improved mooring coverage since 2021 (Chidichimo et al., 2023), but data recorded after 2017 have yet to be incorporated into published AMOC or MHT estimates."

And we now mention in the Conclusions how the expanded dataset could help to determine the cause of differences between estimates:

"Use of the expanded set of moorings will also allow us to determine the importance of aliasing of variability on the boundaries. Since the reanalyses are in reasonable agreement with altimeter-based estimates but not with SAMBA, it prompts closer inspection of the methodologies used to make the computations."

Line 66-77: I think the author's ought to be more careful about lumping all observation based estimates into two simple categories – satellites and in-situ, because some of the latter are based on observations of transport, some on observations of T&S, some on T with other assumptions to get at geostrophy, and some of completely different assumptions to estimate MOC/MHT from floats.

We are not sure we fully understand this comment. We state that the altimeter-based estimate uses both satellite and in-situ data, but we do not separate these categories (i.e., both are used in this estimate) or define exactly what is included in each category.

I realize it is a recent publication and does not include a line at 34.5°S, but I would recommend including a reference to Cainzos et al (2022) in this paragraph as it represents a recent investigation using another prevalent means of estimating the MOC and MHT from in situ observations. https://agupubs.onlinelibrary.wiley.com/doi/full/10.1029/2021GL096527 Authors' choice of course.

Thanks for the suggestion, this is an interesting read and a useful addition to the overview of estimates. We have added this reference to **line 69**.

Line 103: Is there a way to consider how biased all these reanalysis products are to surface observations. That is, how important are subsurface observations to their estimates? I ask because it concerns me that the near surface reanalysis product may be much improved compared to the model alone, but can the same be said for the deep ocean?

We agree that errors in the deeper ocean properties of the reanalysis might be expected to be larger than the upper ocean since there is lower observational coverage in the deep ocean. However, we do not believe this necessarily biases the reanalysis more than a free running model. This is supported by Mignac et al., 2018, where ocean reanalyses are found to have significant improvements in the South Atlantic MOC and heat transports compared to free running models. Since the MOC estimates are integrated from the surface to the ocean floor, the errors below ~1500 m should not impact our MOC estimates.  Fig 1 shows the magnitude of biases in the deep ocean, assuming SAMBA and the altimeter estimate are closer to the truth at this depth. The heat transports could have larger biases since the deep ocean temperature and velocities are used in the MHT calculation. However, a comparison of the temperature and salinity in the GloRan reanalysis against EN4 observations averaged over 1993-2019 in the Atlantic (see figure below) suggests that biases in the time-mean are small in the deep ocean. We also note that these variables have low means and variability in the deep ocean, so the MHT in the deep ocean would be small as well as its variability. We do acknowledge that ocean reanalyses are not perfect, and that there are very few observations in the deep ocean to constrain the reanalyses and evaluate improvements relative to free running models. The increasing number of T, S profiles from South Atlantic moored arrays and cruises, as well as recent deployments of deep Argo floats (Johnson et al., 2020) in the region, will improve deep T, S data coverage.

We note that ocean reanalyses are less accurate at depth in lines 163-164: "The profiles diverge in the deeper ocean, with a weaker southward flow (i.e., the overturning decreases more gradually with depth) and a weaker lower overturning cell in the ensemble than observed. The reanalyses are less accurate at depth due to fewer observations to constrain the flow."

We also note that the reanalyses have been shown to provide close approximations to the AMOC and MHT at the RAPID and OSNAP arrays (Jackson et al., 2019; Baker et al., 2022). This provides confidence that reanalyses are a useful tool to infer the MOC and MHT and

that biases at depth do not have a significant impact on these estimates, at least at these latitudes.

[Figure]

Figure: (Top) Atlantic temperature and salinity fields in GloRanV14 and (lower) differences in these fields between GloRanV14 and EN4 observations, averaged over 1993-2019.

Line 124: Wouldn't one expect a net southward transport in the Atlantic? I would have thought the net zero constraint should be applied globally.

Yes, the net southward transport through the section may well be expected since there is an estimated net transport from the Pacific into the Arctic of ~1.0 Sv (Woodgate, 2018). However, the flow through the Bering Strait is nearly cancelled by the evaporation-minus-precipitation integrated over the entire Atlantic Basin (Lumpkin and Speer, 2007). Given that, mass conservation requires that the transport integrated over the full depth of the ocean across 34.5°S must be close to zero.

It is common to apply a net zero mass or volume transport constraint when calculating the overturning streamfunction (e.g., at RAPID), since the net transport is not physically related to an overturning. However, this could in part cause deviations between the ensemble and the altimeter-based estimate vs. the SAMBA estimate since the latter does not use a similar constraint. We find the net zero volume transport constraint has a small but significant impact on the MOC estimate derived from the reanalyses based on a comparison of estimates from the reanalyses calculated with and without applying the net zero volume constraint (see bracketed values in Table 2 and our response to Lines 119-130 of reviewer 1 for further details). The net zero volume constraint must be applied for the MHT to allow for an estimate in the expected units of heat transport (i.e., Watts). This is consistent with SAMBA (which assumes the volume transport below ~4700 m balances the transport above in the MHT estimate) and the altimeter-based estimate. It also enables comparison with the energy-budget estimate. We choose to use the net zero transport constraint for the MOC

estimates since this is commonly applied when analysing the MOC (Frajka-Williams et al., 2019; Kanzow et al., 2007; McCarthy et al., 2015; Perez et al., 2011), including in the in situ-altimeter estimate examined here (Dong et al., 2021).

And GLORYS12V1??

We now add these values in brackets to Table 2.

Line 127-128: "Each observational product applies its own constraint to reference the flow." This may be true, but I don't understand the intention of this statement here.

We now suggest that the method used to reference the flow may cause some of the differences between the datasets since the barotropic components in the ensemble and SAMBA differ and the altimeter-based estimate has no contribution to the variability from this component. We now concisely describe what these differences are, which is useful to the later discussion.

"Each observational product applies its own constraint to reference the flow due to differences in their geostrophic techniques. The altimeter-based dataset references the flow to the time-mean YoMaHA at 1000 m and uses a zero net mass transport constraint (Dong et al., 2021). Kersalé et al. (2020) use models to reference the time-mean barotropic component at 1500 db, and bottom pressure measurements from the moorings provide the time-varying barotropic velocity component at the common depth between adjacent moorings."

Figure 1: Where is AABW?

AABW is the water mass below depths of ~4200 m where the volume transport increases with depth towards 0 Sv (i.e., a northward flow) (see Fig 1). It is very weak in the reanalyses below ~4000 m and non-existent in density space (since there is no abyssal cell) as we note in lines 161-163:
"The overturning profiles of the ensemble and GLORYS12V1 in density space have a stronger maximum overturning transport than that calculated in depth space, and they have no negative transport (i.e., no abyssal cell) (Fig. 1b)."

Line 144: ...due to the existence of fewer observations ...  (is this discussed elsewhere)

 Thanks, we now say: "due to there being fewer observations to constrain the flow".

Line 146-147: density space versus pressure space. I completely agree, but this is not new observation nor is it specific to 34.5°S. Take a look, for example, at Lumpkin and Speer (2007) or the many works of Susan Lozier.

We agree this is not a new observation and we do not think it is necessary to go into details on the overturning pathways since it is not particularly relevant to this study. We have therefore removed this statement.

Line 148-149 – It seems this is an important point. Is there no way to include an illustration of this lack of sensitivity, even if it is shown as supplementary material?

We have added a supplementary figure (Fig S1) comparing the monthly-mean timeseries of the ensemble mean calculated in depth and density space along 34.5S over the whole 1993-2020 period and over the SAMBA observational period.

Line 155: Perhaps I am misunderstanding. Given the overlap, I don't see how one can make any comparison at all amongst these products. I also don't understand how one can talk about an "increased spread in the time-mean MHT compared to the MOC." What scale is being used?

Given the uncertainty, it is true that the observed time-mean MOC across SAMBA could be the same as the other estimates. Hence, we state "These ranges are within the documented uncertainty of SAMBA (Table 2)". All observations have uncertainties associated with them due to measurement errors, although those associated with SAMBA are larger than the other estimates. Nonetheless, we believe it is useful to compare the time-mean values between the different datasets since these are the best estimates of each dataset. We note that time-mean biases should not impact the variability (i.e., the anomalies from the time-mean).

We mean there is greater spread between the datasets in the time-mean MHT than in the time-mean MOC, relative to the ensemble mean value. The MHT has a 75% increase from the minimum to the maximum values (0.61 PW – 0.36 PW) relative the ensemble-mean of 0.36 PW, whereas the MOC has only a 20% increase (18.72 Sv – 16.56 Sv) relative to the ensemble-mean of 16.56 Sv. We agree this was confusing, so we now make clear this is relative to the ensemble-mean value. Even though they are different quantities, a 75% higher MHT would be considered large whereas a 20% increase in the AMOC is more reasonable. We now write this more clearly by making clearer what the changes are relative to, in lines 173-176.

Line 170: Again, I feel like I am missing something here. Don't the ensemble means have the satellite record embedded in their estimates? If so, it would indicate a major issue if the two were not strongly related. Okay, I see the discussion Lines 180-184, but I would still argue that reanalyses are so dependent of the satellite record and that their characteristics must by necessity be closer to the near-surface focused observations. If this is the case, perhaps lines 180-184 should be the lead-in to the discussion.

Yes, both datasets use some of the same satellite observations (the temperature profiles) and thus perhaps they are expected to be closer to the altimeter-based estimate than SAMBA. However, the methods used differ substantially, so their similarity provides support for these methods. Nonetheless, there are still significant differences in the timeseries and their seasonal cycles. We cannot say which of the three estimates is most reflective of reality so comparing them in full to understand the similarities and differences is crucial. The reanalysis uses observations of SST and T/S profiles that are also used by altimeter-based estimate (in addition to the SLA observations that we have shown are not responsible for the high correlation).

We now comment in lines 194-196 that the similarity between these estimates could be caused by both estimates using some of the same observational datasets in their implementation:
"The experimental reanalysis does, however, still assimilate in-situ and satellite temperature and salinity data, which would serve to constrain thermosteric and halosteric, respectively, contributions to sea level."

Lines 192-195 – Perhaps there are too many curves or the figure is too squashed, I cannot see these trends in Figure 3. The same is true for the trends noted in the next paragraph.

We agree it is difficult to see many of these trends. This is primarily because the trends are small, so they are difficult to see by eye without a very stretched y-axis. The SAMBA data also has a much greater range than the other datasets so the y-axis must have a greater range. GloRanV14 has the largest MOC trend which can be seen with close inspection (i.e., low values prior to 2006 and generally higher values thereafter). The increase in the altimeter-based estimate and the energy-budget MHT estimates can also be seen. The other trends are too small to make out on these plots. We have increased the y-dimension of each of these plots so the change in values is more apparent, and we have added trend information to Table 2.

Lines 205-210 – which begs the questions are the models and satellite estimates getting the boundary currents and is there something important in the middle of the basin that SAMBA is missing?

This is an interesting point. The reanalyses and altimeter-based estimate could potentially be misrepresenting the boundary currents and their contributions to the MOC variability. We note that while altimetric SLA agree well with dynamic height anomalies at the interior SAMBA sites (Kersale et al. 2021), they don't agree as well at the inshore moorings where both baroclinic and barotropic variability are important. The reanalyses have higher resolution than SAMBA across the whole section, so SAMBA could also be over-estimating/aliasing variability at the boundaries. On the other hand, SAMBA could be directly measuring barotropic variability that the reanalyses and altimeter-based estimates are not capturing on the boundaries.

Since the baroclinic component of the MOC largely determines the phase of the geostrophic seasonal cycles, we analyse the seasonal variability of the baroclinic velocity across the basin in the reanalyses. We find the boundary currents have significant seasonal variability in the reanalyses, with both the eastern and western boundaries contributing significantly to the total variability. The contribution to the seasonality from seasonal differences in the western boundary transport dominates over 1993-2020. However, over 2013-2017, seasonal differences in the interior baroclinic transport are also significant, so may play a role. We discuss these spatial contributions to the seasonal cycle in lines 287-293.

We now provide suggestions for future work (lines 343-356) including analysing the spatial variation of the SAMBA estimate by comparing estimates between each profile and the T/S

profiles themselves with the reanalysis. This could help understand where the SAMBA estimates differ from the other datasets in both the time-mean values and in the variability. We suggest that finer horizontal resolution of the datasets over the sloping topography of the continental shelves may be critical for resolving well the barotropic and baroclinic variations on the western and eastern boundaries. There was a drop in the number of locations for which data was being collected at SAMBA moorings from the middle of 2017 – 2021 (see our response to reviewer 1 for further details). The number of active moorings has increased since then, and this will allow us to examine where the nine-site SAMBA calculation may have aliased variability on the boundaries. This is beyond the scope of this paper.

Line 274-279 – These sentences come off as an excuse not provide something thinking and analysis into the causes of differences or similarities. The true impact will come with some attempt to answer these questions.

Thanks for making this point. We agree this paper did not explain the differences between the estimates. We now compare the baroclinic and barotropic components in the ensemble with SAMBA to narrow down the cause of the differences. We discuss these differences in lines 263-313, showing that the phase and magnitude of the seasonal cycle of the baroclinic MOC component differs in SAMBA and the ensemble.

We believe the baroclinic component is crucial since it primarily determines the shape of the seasonal cycles, which differ between the datasets. This also causes some of the differences in the magnitude of the monthly-mean and inter-annual variability since the baroclinic and barotropic components tend to oppose each other in the ensemble, but augment each other in SAMBA.

Differences may also be due to the way that the baroclinic flow is referenced in the reanalyses (i.e., zero net mass transport) vs. in the observations (i.e., the use of a reference velocity). The observations take different approaches to referencing the flow and this may be yet another source of difference. This is an important subject for future analysis. Our analysis narrows down the possible causes of the difference – and will lead to potential improvements to the referencing of the flow in the SAMBA and in situ-altimetry estimates bringing the estimates that result from the two different methodologies into better agreement. It also shows that reanalysis are a useful tool to identify, characterize, and understand the differences, pointing both the observational and modelling communities towards improved methodologies.

We then suggest further work that could be done to determine the cause of the differences in lines 343-356. The vertical profiles of density (and temperature and salinity) in the reanalyses, SAMBA estimate, and altimeter-based estimate could be directly compared to determine how and if they differ. Also, estimates of the baroclinic component using different density profiles with different zonal spacing could be used in the reanalyses to infer the impact of the different resolution of the datasets. Analysis using the vertical density profiles may show the cause of differences in the long-term mean, monthly, and inter-annual variability and seasonal cycles between estimates. Similarly, the reanalyses velocity can be used to estimate the barotropic velocity (vertically averaged) along 34.5S, which can then be compared with the reference velocity used by the in situ-altimetry and

SAMBA methods to reference the flow. Analysis of the two-dimensional fields of baroclinic and barotropic velocity along 34.5S would allow spatial differences in the velocity fields between estimates to be inferred, enabling further understanding of the cause of differences. Specifically, we could examine how they differ on the boundaries over sharply sloping topography and in the interior. We view this as an important next step to take in future studies. The need for delving into these details has been heightened as a result of model-observational comparisons such as the one conducted in this study.

Trivia:
Line 190: … running mean overturn and MHT … **are** relatively…
And elsewhere - "overturn" is the noun, "overturning" is the adjective

Thanks, we have changed "is" to "are"

We have chosen to keep the text as "overturning" since this is used to refer to the "thing" i.e., the meridional overturning circulation, throughout the literature. We note also after some research that it could be considered to be a gerund (a word based on a verb and expressing an action but serving another function, in this case a subject).

Table 1, Figures 1 & 2 and text:  The SAMBA and Dong estimates. It took me a while to find the statement (lines 104-106 stating whose estimates were being used.  These references should also be cited in at least one of the figure captions and in the succeeding captions the reader should be referred back to the figure that includes the citations.

Thanks, we have now added the references to Figure 1 and refer to this figure in the succeeding figures. Table 2 has the product reference numbers added in brackets.

**References**

Dong, S., Goni, G., Domingues, R., Bringas, F., Goes, M., Christophersen, J., & Baringer, M. (2021). Synergy of In Situ and Satellite Ocean Observations in Determining Meridional Heat Transport in the Atlantic Ocean. *Journal of Geophysical Research: Oceans*, *126*(4), e2020JC017073. https://doi.org/10.1029/2020JC017073

Frajka-Williams, E., Ansorge, I. J., Baehr, J., Bryden, H. L., Chidichimo, M. P., Cunningham, S. A., Danabasoglu, G., Dong, S., Donohue, K. A., Elipot, S., Heimbach, P., Holliday, N. P., Hummels, R., Jackson, L. C., Karstensen, J., Lankhorst, M., le Bras, I. A., Susan Lozier, M., McDonagh, E. L., … Wilson, C. (2019). Atlantic meridional overturning circulation: Observed transport and variability. *Frontiers in Marine Science*, *6*(JUN), 260. https://doi.org/10.3389/FMARS.2019.00260/BIBTEX

Jackson, L. C., Dubois, C., Forget, G., Haines, K., Harrison, M., Iovino, D., Köhl, A., Mignac, D., Masina, S., Peterson, K. A., Piecuch, C. G., Roberts, C. D., Robson, J., Storto, A., Toyoda, T., Valdivieso, M., Wilson, C., Wang, Y., & Zuo, H. (2019). The Mean State and Variability of the North Atlantic Circulation: A Perspective From Ocean Reanalyses. *Journal of Geophysical Research: Oceans*, *124*(12), 9141–9170. https://doi.org/10.1029/2019JC015210

Johnson, G. C., Cadot, C., Lyman, J. M., McTaggart, K. E., & Steffen, E. L. (2020). Antarctic Bottom Water Warming in the Brazil Basin: 1990s Through 2020, From WOCE to Deep Argo. *Geophysical Research Letters*, *47*(18), e2020GL089191. https://doi.org/10.1029/2020GL089191

Kanzow, T., Cunningham, S. A., Rayner, D., J-M Hirschi, J., Johns, W. E., Baringer, M. O., Bryden, H. L., Beal, L. M., Meinen, C. S., & Marotzke, J. (2007). *Supporting Online Material Observed Flow Compensation Associated with the MOC at 26.5°N in the Atlantic*. https://doi.org/10.1126/science.1141304

Kersalé, M., Meinen, C. S., Perez, R. C., le Hénaff, M., Valla, D., Lamont, T., Sato, O. T., Dong, S., Terre, T., van Caspel, M., Chidichimo, M. P., van den Berg, M., Speich, S., Piola, A. R., Campos, E. J. D., Ansorge, I., Volkov, D. L., Lumpkin, R., & Garzoli, S. L. (2020). Highly variable upper and abyssal overturning cells in the South Atlantic. *Science Advances*, *6*(32). https://doi.org/10.1126/SCIADV.ABA7573/SUPPL_FILE/ABA7573_SM.PDF

Kersalé, M., Meinen, C. S., Perez, R. C., Piola, A. R., Speich, S., Campos, E. J. D., Garzoli, S. L., Ansorge, I., Volkov, D. L., le Hénaff, M., Dong, S., Lamont, T., Sato, O. T., & van den Berg, M. (2021). Multi-Year Estimates of Daily Heat Transport by the Atlantic Meridional Overturning Circulation at 34.5°S. *Journal of Geophysical Research: Oceans*, *126*(5), e2020JC016947. https://doi.org/10.1029/2020JC016947

Lin, X., & Munday, D. R. (n.d.). *Southern Ocean Wind Stress in CMIP5 Models: Role of Wind Fluctuations*. https://doi.org/10.1175/JCLI-D-19-0466.1

McCarthy, G. D., Smeed, D. A., Johns, W. E., Frajka-Williams, E., Moat, B. I., Rayner, D., Baringer, M. O., Meinen, C. S., Collins, J., & Bryden, H. L. (2015). Measuring the Atlantic Meridional Overturning Circulation at 26°N. *Progress in Oceanography*, *130*, 91–111. https://doi.org/10.1016/J.POCEAN.2014.10.006

Meinen, C. S., Speich, S., Piola, A. R., Ansorge, I., Campos, E., Kersalé, M., Terre, T., Chidichimo, M. P., Lamont, T., Sato, O. T., Perez, R. C., Valla, D., van den Berg, M., le Hénaff, M., Dong, S., & Garzoli, S. L. (2018). Meridional Overturning Circulation Transport Variability at 34.5°S During 2009–2017: Baroclinic and Barotropic Flows and the Dueling Influence of the Boundaries. *Geophysical Research Letters*, *45*(9), 4180–4188. https://doi.org/10.1029/2018GL077408

Mignac, D., Ferreira, D., & Haines, K. (2018). South Atlantic meridional transports from NEMO-based simulations and reanalyses. *Ocean Science*, *14*(1), 53–68. https://doi.org/10.5194/OS-14-53-2018

Perez, R. C., Garzoli, S. L., Meinen, C. S., & Matano, R. P. (2011). Geostrophic Velocity Measurement Techniques for the Meridional Overturning Circulation and Meridional Heat Transport in the South Atlantic. *Journal of Atmospheric and Oceanic Technology*, *28*(11), 1504–1521. https://doi.org/10.1175/JTECH-D-11-00058.1

Woodgate, R. A. (2018). Increases in the Pacific inflow to the Arctic from 1990 to 2015, and insights into seasonal trends and driving mechanisms from year-round Bering Strait mooring data. *Progress in Oceanography*, *160*, 124–154. https://doi.org/10.1016/J.POCEAN.2017.12.007

Baker J, Renshaw R, Jackson L, Dubois C, Iovino D, Zuo H. (2022). Overturning variations in the subpolar North Atlantic in an ocean reanalyses ensemble. Copernicus Marine Service Ocean State

Report, Issue 6, Journal of Operational Oceanography.

---

## Author Response (AR2)

**Response to reviewer comments for manuscript: "South Atlantic overturning and heat transport variations in an ocean reanalysis ensemble and observation-based estimates"**

I thank the authors for addressing my previous comments carefully. The revised manuscript has improved and the new detailed comparisons between reanalyses and observation-based estimates are helpful. I recommend the manuscript for publication after the following minor points are addressed.

We are happy our changes have satisfied the reviewer's main comments and that this has improved the paper. We thank them greatly for their valuable feedback and for the additional comments they have provided below that help improve the paper's clarity. We respond to their comments and questions below in blue font.

Title: It is awkward to include 'other estimates' in the title. Is it referring to those 'observation-based estimates'? Please rephrase and it would be better to be specific.

Thanks, we have changed the title to "observation-based estimates" to make it clear we are referring to estimates of the MOC and MHT and to be more specific. We have not included details of the specific observations used to prevent the title becoming too long.

Lines 141-146: What velocity is used at the reference level depth? Is it assumed to be zero all the time? Is it the time-mean or time-varying velocity taken from the model at that reference level depth? Some details on this calculation are currently missing, making it hard to understand how the authors obtain their baroclinic velocities (and thus the baroclinic component of the MOC).

When calculating the baroclinic MOC anomaly in the reanalyses, we set the velocity at the reference level depth to zero throughout the timeseries and calculate the baroclinic velocity above this level using the thermal wind balance equations. We make this choice because we use these velocity estimates to calculate the baroclinic MOC anomalies, so we do not include the barotropic velocity in this calculation. Since we calculate anomalies of the baroclinic MOC from their time-mean values in each reanalysis rather than the absolute magnitude of the baroclinic MOC, our estimate would be the same if we set the velocity at the reference level depth to the time-mean velocity at this depth.

We now make this important detail explicit in the methods by adding a sentence to lines 146-147: "The reference velocity is not required to calculate the baroclinic MOC anomalies, so we set the baroclinic velocity to zero at the reference level depth"

Lines 151-155: I cannot follow the justification of the reanalyses-SAMBA comparisons given their known incompatibility.

We agree this sentence was confusing. The reanalyses and SAMBA estimates do not use the same reference levels, so some of the differences between the estimates of the baroclinic and barotropic MOC anomalies could arise from the different reference levels. However, given both estimates have reference levels at depths below the depths over which the baroclinic velocity varies greatly (at least based on the velocities in the reanalyses), most of the differences between their baroclinic and barotropic MOC anomaly estimates are likely due to differences in the geostrophic MOC (i.e., what we are trying to understand) rather than differences in the reference levels. The reanalysis estimate of the baroclinic MOC anomaly is also not very sensitive to the choice of reference level. We note these points in the method section.

We think adding "and methodologies" to this sentence was confusing because the fact the methodology (besides the reference levels) is different does not mean comparisons cannot be made between the estimates. Instead, the methodology differences likely cause some of the differences in the geostrophic MOC that we are trying to understand. We have now removed "and methodologies", which hopefully makes this clearer.

Lines 263-266: Same question as above. Their incompatibility seems to make the objective of this section impossible (that is to 'investigate possible causes of the differences in variability between SAMBA and the ensemble'). Please elaborate.

Thanks, we agree this was also confusing and made the comparison seem inappropriate.

As above, we now only mention the reference levels being different. We remove "and method of computing the reference level velocity; nonetheless, major features can be inferred from each dataset." We now state in lines 264-266:

"The baroclinic and barotropic components of the MOC are not directly comparable between the ensemble and SAMBA due to differences in the reference level depth, but this probably has little impact on the differences between these estimates (see Section 2.2)."

While the method of computing the reference level velocity (and other differences in the methodologies) cause some of the differences in the baroclinic and barotropic MOC anomalies between the estimates, they also cause differences between the estimates in the geostrophic MOC that we are trying to understand. Thus, the impact of these differences on the MOC anomalies should be included in our estimates of the baroclinic and barotropic MOC anomalies. While the different reference level depths prevent direct comparison between these estimates, they likely only cause small differences between the estimates as described in the previous comment.